# Discovering Nonlinear PDEs from Scarce Data with Physics-encoded Learning

**Chengping Rao**[†], **Pu Ren**[†]**, Yang Liu**
Northeastern University
{rao.che, ren.pu, yang1.liu}@northeastern.edu

**Hao Sun**[*]
Renmin University of China
haosun@ruc.edu.cn

## Abstract

There have been growing interests in leveraging experimental measurements to discover the underlying partial differential equations (PDEs) that govern complex physical phenomena. Although past research attempts have achieved great success in data-driven PDE discovery, the robustness of the existing methods cannot be guaranteed when dealing with low-quality measurement data. To overcome this challenge, we propose a novel physics-encoded discrete learning framework for discovering spatiotemporal PDEs from scarce and noisy data. The general idea is to (1) firstly introduce a novel deep convolutional-recurrent network, which can encode prior physics knowledge (e.g., known PDE terms, assumed PDE structure, initial/boundary conditions, etc.) while remaining flexible on representation capability, to accurately reconstruct high-fidelity data, and (2) perform sparse regression with the reconstructed data to identify the explicit form of the governing PDEs. We validate our method on three nonlinear PDE systems. The effectiveness and superiority of the proposed method over baseline models are demonstrated.

## 1 Introduction

Deriving physical laws remains critical for understanding the dynamical patterns and complex spatiotemporal behaviors in nature. Well formulated governing equations facilitate scientific discovery and promote establishment of new disciplines. Traditionally, scientists seek to find governing equations by rigorously following the first principles, including the conservation laws, geometric brownian motion assumptions and knowledge-based deductions. However, these classical approaches show limited capability and slow progress in identifying clear analytical governing equations, and leave numerous complex systems under-explored. Thanks to the recent achievements in machine learning (ML), advances in governing equation/law discovery has been drastically accelerated.

The pioneer works (Bongard & Lipson, 2007; Schmidt & Lipson, 2009) apply symbolic regression to reveal the underlying differential equations that govern nonlinear dynamical systems without any prior background knowledge. Although this inspiring investigation implies the dawn of discovering the fundamental principles automatically in scientific ML, the scalability and overfitting issues herein have been the critical concerns. Recently, a ground-breaking work by (Brunton et al., 2016) introduces the sparse regression into the discovery of parsimonious governing equations (aka., Sparse Identification of Nonlinear Dynamics (SINDy)), based on the assumption that the dynamical systems are essentially controlled by only few dominant terms, which evolves to promote the sparsity in the candidate functions. Furthermore, the introduction of spatial derivative terms and the improvement of the sparsity-promoting algorithm as Sequential Threshold Ridge regression (STRidge) make SINDy applicable to general PDE discovery (i.e., the PDE-FIND algorithm) (Rudy et al., 2017). However, due to the demanding requirement of derivative estimation via numerical differentiation, these sparse regression methods largely rely on high-quality measurement data, which is barely accessible in the real-world sensing environment. Therefore, there is an urgent need to address the issue of discovering the underlying governing equations from low-resolution (LR) and noisy measurement data. Very recently, many works attempt to relieve the pain by reconstructing the high-resolution (HR) scientific data from LR measurements and then discovering the governing laws, including robust low-rank discovery (Li et al., 2020a), physics-informed spline learning (Sun et al., 2021) and physics-informed deep learning (Chen et al., 2021). Nevertheless, the dominant

---

[*]Corresponding author      [†]Equal contribution

reconstruction part lacks explicit physical interpretability (i.e., grey-box model), which may induce the difficulty in optimization especially for high-dimensional PDE systems with delicate patterns.

**Contributions.** To overcome the challenges resulting from the noisy and LR measurements, we propose a novel physics-encoded deep learning framework (i.e., white-box model) for discovering the governing equations of physical systems. The main contributions of this paper are three-fold:

- We design a novel recurrent network architecture that is able to fit spatiotemporal measurement data for HR metadata generation with high accuracy and physical consistency. Notably, this network is characterized with capability to encode given physics knowledge (e.g., known terms, assumed PDE structure based on multiplicative feature operation, initial/boundary conditions) so that the given physics as model prior is respected rigorously.

- Based on the proposed reconstruction network, we develop a hybrid optimization approach to discover spatiotemporal PDEs from noisy and LR measurements. Specifically, to determine the explicit form of underlying governing equations, sparse regression is performed by using HR prediction inferred from the model. Finally, through inheriting the PDE terms and coefficients from the sparse discovery, we conduct the fine-tuning step with the completely physics-based network to further improve the accuracy of the scientific discovery.

- The proposed coupled framework establishes the physics-encoded data-driven predictive model from the noisy and LR measurement. To evaluate the performance of the model, we perform three numerical experiments on a variety of nonlinear systems. It is found that the proposed approach demonstrates excellent robustness and accuracy in spite of the high level of noise and the scarcity (spatially and temporally) in the measurements.

## 2 RELATED WORK

**Dynamical system modeling.** Simulating multi-dimensional PDE systems with neural networks (NNs) has been a renewed research topic, which can be dated back to last century (Lee & Kang, 1990; Lagaris et al., 1998). Like the traditional simulation methods, the recent breakthrough in NN-based modeling for physical systems also fall into two streams: continuous and discrete learning. The continuous learning scheme can avoid the notorious demand of time-step and grid sizes for numerical stability. Among the meshfree strategies, based on the incorporation of prior knowledge of PDE systems, there exist two directions seeking for the approximation of simulation with DNNs, i.e., the pure data-driven approaches (Han et al., 2018; Long et al., 2018; Wang et al., 2020) and physics-informed learning (Yu et al., 2017; Raissi et al., 2019; Rao et al., 2021b; Karniadakis et al., 2021). The former option relies on large amounts of high-quality data, while the physics-informed ML only requires scarce or even no labeled data due to the enhancement from physical constraints. Besides, it is worthwhile to mention that the recent studies on neural operators (Li et al., 2020b; Lu et al., 2021; Patel et al., 2021) have exhibited the great potential of learning the nonlinear, meshfree and infinite-dimensional mapping with DNNs for physical systems.

Regarding the discrete learning scheme, the mesh-based methods play a dominant role in simulating complex high-dimensional PDE systems. Herein, the convolutional architectures are mainly applied for handling regular grids in steady-state problems (Zhu & Zabaras, 2018; Zhu et al., 2019) and spatiotemporal systems (Bar-Sinai et al., 2019; Geneva & Zabaras, 2020; Kochkov et al., 2021; Ren et al., 2021), as well as irregular domains by considering elliptic coordinate mapping (Gao et al., 2021a). Moreover, with the theoretical development of graph neural networks (GNNs) (Kipf & Welling, 2016; Bronstein et al., 2017; Battaglia et al., 2018), researchers seek for a new direction for geometry-adaptative learning of nonlinear PDEs with arbitrary domains (Belbute-Peres et al., 2020; Sanchez-Gonzalez et al., 2020a; Gao et al., 2021c). In addition to the common mesh-based approaches, the particle-based frameworks have been attracting a lot of interest in scientific modeling for physical systems, especially by employing GNNs (Li et al., 2018; Ummenhofer et al., 2019; Sanchez-Gonzalez et al., 2020b).

**Scientific data augmentation.** Scientific data are typically sparse, noisy and incomplete. Therefore, data augmentation has been an active research area and received massive attention in the scientific ML community, e.g., data reconstruction from sparse measurements (Raissi, 2018; Callaham et al., 2019; Erichson et al., 2020; Rao et al., 2021a), super-resolution (SR) (Stengel et al., 2020; Gao et al., 2021b) and denoising (Fathi et al., 2018). Generally, based on the type of incorporated

data, we categorize the existing data augmentation methods into two groups. The first one is to extract the coherent structure and correlation features as the basis by utilizing the sufficient high-quality datasets. For instance, in the context of SR, the majority of NN-based frameworks (Liu et al., 2020; Esmaeilzadeh et al., 2020; Fukami et al., 2021) aim to reconstruct the high-resolution (HR) full-field data from low-resolution (LR) measurements by employing HR ground truth reference as the constraint. Nevertheless, the second type of data augmentation takes advantage of the physics-based model and can accurately capture the high-fidelity dynamical patterns from the LR and noisy measurement data. The recent advances of physics-based learning models have shown the the great success of data augmentation for PDE systems with only sparse and noisy data (Raissi, 2018; Rao et al., 2021a) and even without any labeled data (i.e., only with PDEs) (Gao et al., 2021b).

**Data-driven discovery.** The data-driven discovery is also a renascent research attempt. The earliest work (Dzeroski & Todorovski, 1995) introduced the inductive logic programming to find the natural laws from experimental data. With the rapid advance of machine intelligence, the past three decades have seen a considerable amount of literatures growing up around the interplay between ML and scientific discovery. Essentially, two genres have been observed in distilling governing equations, with respect to the searching of candidate functions. Firstly, the natural and optimal solution to automatically identify the governing equations for dynamical systems is to learn a symbolic model from experimental data. Hence, symbolic regression (Schmidt & Lipson, 2009) and symbolic neural networks (SNN) (Sahoo et al., 2018; Kim et al., 2020) have been explored for inferring the physical models without any prior knowledge. In addition, GNN (Cranmer et al., 2020) has also been studied to identify the nontrivial relation due to its excellent inductive capability.

The second group manually defines a large library functions based on the prior knowledge, and utilizes the sparse-prompting algorithms to discover the parsimonious physics model efficiently. The most representative works are SINDy (Brunton et al., 2016) and PDE-FIND (Rudy et al., 2017) for ordinary differential equations (ODEs) and PDEs, respectively. Nevertheless, the standard sparse representation-based methods are limited to the high-fidelity noiseless measurement, which is usually difficult and expensive to obtain. Hence, many recent efforts have been devoted to discover PDEs from sparse/noisy data (Li et al., 2020a; Xu et al., 2021; Sun et al., 2021; Chen et al., 2021).

## 3 METHODOLOGY

### 3.1 PROBLEM DESCRIPTION

Discovering the explicit form of the governing equation(s) for a given system from the measurement data is of critical importance for scientists to understand some underexplored processes. To formalize this problem, which is also known as the data-driven equation discovery, let us assume the response of a multi-dimensional spatiotemporal system is described by

$$\mathbf{u}_t = \mathcal{F}\left(\mathbf{x}, t, \mathbf{u}, \mathbf{u}, \mathbf{u}_x, \mathbf{u}_y, \nabla\mathbf{u}, \cdots\right) \qquad (1)$$

where $\mathbf{x} = [x, y]^{\mathrm{T}}$ and $t$ denote the spatial and temporal coordinates respectively, $\mathbf{u}$ denotes the state variable possibly consisting of multiple components (e.g., $\mathbf{u}$ has two components $u$ and $v$) and the subscript denotes the partial derivatives, e.g., $\mathbf{u}_x = \partial\mathbf{u}/\partial x$. $\mathcal{F}$ is a function parameterizing the system dynamics with possible linear/nonlinear terms (e.g., $\mathbf{u}^2$, $\mathbf{u}_x$ and $\nabla\mathbf{u}$, where the Nabla operator $\nabla$ is defined as $[\partial/\partial x, \partial/\partial y]^{\mathrm{T}}$). The objective of the data-driven equation discovery is to uncover the closed form of $\mathcal{F}$ provided time series measurements on some fixed spatial locations. Although this problem has been studied extensively over the past few years, the combinatorially large search space of the possible equation appears as one of the major obstacles that hinders the discovery of the governing equation from very limited measurements. The existence of noise, as well as the coarse-sampling of the measurement, also adds to the difficulty of this problem. To address these challenges, we proposed a novel physics-encoded recurrent network that is able to reconstruct the high-fidelity solution from the noisy and LR measurements with the help of prior physics knowledge. We successfully combine this network with the sparse regression for discovering the explicit form of the governing PDE of dynamical systems.

### 3.2 PHYSICS-ENCODED NETWORK FOR DATA RECONSTRUCTION

Obtaining the numerical derivatives for constructing the library matrix is crucial to the equation discovery with sparse regression. Due to the LR and the noise in measurements, directly applying

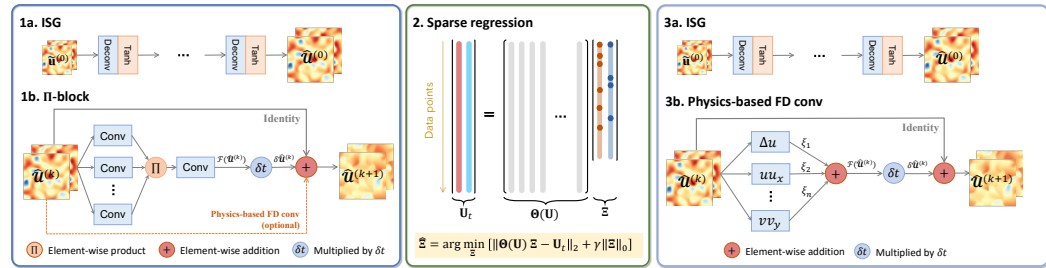

Figure 1: Discovery of governing PDEs with the proposed Physics-encoded DL framework. **1. Data reconstruction:** data-driven model constructed from noisy and LR measurement is used to generate HR solution for sparse regression; **2. Sparse regression:** STRidge algorithm is used to obtain the sparse coefficient vector $\Xi$; **3. Coefficients fine tuning:** the PeRCNN built based on identified PDE structure is employed to fine-tune the coefficient from the sparse regression.

the traditional finite difference (FD) onto the raw measurements may lead to inaccurate derivatives, which would significantly complicate the equation discovery. To handle this issue, we introduce a recurrent network architecture, namely the Physics-encoded Recurrent Convolutional Neural Network (PeRCNN) shown in Fig. 1(1a) and (1b), to establish the data-driven predictive model from LR noisy measurements. PeRCNN consists of a fully convolutional (Conv) network as initial state generator (ISG) and a well designed Conv block, namely $\Pi$-block (product) for updating the state variables on high-resolution grid recurrently. Note that our network does not require any HR data as the "label" for training, which is fundamentally different from the classic super resolution task.

The network architecture mimics the forward Euler time stepping $\widehat{\boldsymbol{\mathcal{U}}}^{(k+1)} = \widehat{\boldsymbol{\mathcal{U}}}^{(k)} + \mathcal{F}(\widehat{\boldsymbol{\mathcal{U}}}^{(k)}; \boldsymbol{\theta})\delta t$, where $\widehat{\boldsymbol{\mathcal{U}}}^{(k)} \in \mathbb{R}^{H \times W}$ denotes the HR prediction on a $H \times W$ grid at $k^{\text{th}}$ time step, $\delta t$ is the time spacing and $\boldsymbol{\theta}$ is a set of trainable variables for approximating $\mathcal{F}$. Since we seek the predictive model for HR solution, the ISG component is introduced to generate a HR initial state from the noisy and LR measurement $\tilde{\mathbf{u}}^{(0)}$ for the recurrent computation. The recurrent $\Pi$-block is a major innovation of this network architecture to capture the dynamics of $\mathcal{F}$. As shown in Fig. 1(1b), the input state variable to $\Pi$-block would firstly be mapped to the hidden space through multiple parallel Conv layers, whose output is then fused via the element-wise product operation. A Conv layer with $1 \times 1$ kernel (Lin et al., 2013) is subsequently used to linearly combine multiple channels into the output (i.e., approximated $\mathcal{F}$). Mathematically, the $\Pi$-block seeks to approximate the function $\mathcal{F}$ via polynomial combination of solution $\mathbf{u}$ and its spatial derivatives, given by

$$\widehat{\mathcal{F}}(\widehat{\boldsymbol{\mathcal{U}}}^{(k)}) = \sum_{c=1}^{N_c} f_c \cdot \left[ \prod_{l=1}^{N_l} \left( \mathcal{K}^{(c,l)} \circledast \widehat{\boldsymbol{\mathcal{U}}}^{(k+1)} + b_l \right) \right] \tag{2}$$

where $N_c$ and $N_l$ denote the numbers of channels and parallel Conv layers respectively; $\circledast$ denotes the Conv operation; $(c, l)$ indicates the Conv filter $\mathcal{K}$ of $l$-th layer and $c$-th channel; $f_c$ is the weight in $1 \times 1$ Conv layer while the bias is omitted for simplicity. This multiplicative representation of $\Pi$-block promotes the network expressiveness for nonlinear function compared with the additive representation (i.e., weighted sum of multiple output) commonly seen, such as in PDE-Net (Long et al., 2018). We demonstrate in Appendix Section A that $\Pi$-block is an universal polynomial approximator to the nonlinear function $\mathcal{F}$. In addition, this network architecture features the capability to encode physics knowledge, such as the existing term in the PDE or the known initial and boundary conditions (I/BCs). As shown in Fig. 1(1b), a highway physics-based Conv layer[1] can be created to account for existing terms known *a priori* in the PDEs. In Appendix Section D, we demonstrate that such a highway connection can accelerate the training speed and improve the accuracy of the fitted model. In addition, the knowledge of I/BCs of the system can be encoded into the network through the customized padding in Conv operation. For example, the periodic boundary condition (PBC) could be encoded into the model via circular padding on the prediction (e.g., [1,2,3,4] padded to [4,1,2,3,4,1]). Note that "physics-encoded" has the meaning of threefold: (1) PDE solution time marching that follows the recurrence of forward Euler integration, (2) encoding *a priori* PDE terms into the $\Pi$-block, and (3) encoding multiplicative polynomial-type PDE terms into the $\Pi$-block.

---

[1]This Conv layer with the corresponding FD filter would be frozen throughout the training.

### 3.3 SPARSE DISCOVERY

Based on the reconstructed HR solution (generated metadata), we further employ the sparse discovery algorithm (Rudy et al., 2017) to distill the underlying governing equations. The sparse regression method roots on a critical observation that the right hand side (RHS) of Eq. (1) for the majority of natural systems consists of only a few terms, i.e., the $\mathcal{F}$ demonstrates sparsity in the space of possible functions. Given a library of candidate functions of $\mathcal{F}$, we can formulate the discovery of the governing equation as a regression problem. Let us consider the state variable with one single component $u \in \mathbb{R}^{n_s \times n_t}$, which is defined on $n_s$ spatial locations and within $n_t$ time steps. After flattening the state variable into a column vector $\mathbf{U} \in \mathbb{R}^{n_s \cdot n_t \times 1}$, a library $\boldsymbol{\Theta}(\mathbf{U}) \in \mathbb{R}^{n_s \cdot n_t \times s}$ that encompasses a pool of $s$ candidate functions (e.g., linear, nonlinear terms, derivatives, etc.) can be constructed to represent the governing equations of physical systems. Each column in $\boldsymbol{\Theta}(\mathbf{U})$ represents a candidate function in $\mathcal{F}$, such as $\boldsymbol{\Theta}(\mathbf{U}) = [\mathbf{1}, \mathbf{U}, \mathbf{U}^2, \dots, \mathbf{U}_x, \mathbf{U}_y, \dots]$. Assuming that $\boldsymbol{\Theta}(\mathbf{U})$ has a sufficiently rich column space, the governing equation described by Eq. (1) can be rewritten as a linear system by using a sparse coefficient vector $\boldsymbol{\Xi} \in \mathbb{R}^{s \times 1}$, namely,

$$\mathbf{U}_t = \boldsymbol{\Theta}(\mathbf{U})\boldsymbol{\Xi} \tag{3}$$

To compute a reasonable $\boldsymbol{\Xi}$ such that its sparsity is satisfied while the regression error is small, the Sequential Threshold Ridge regression (STRidge) algorithm (Rudy et al., 2017), among other effective sparsity-promoting methods such as the Iterative Hard Thresholding (IHT) method (Haupt & Nowak, 2006; Blumensath & Davies, 2009), is used in this paper due to its superior performance compared with other sparsity-promoting algorithms, such as LASSO (Tibshirani, 1996) and Sequentially Thresholded Least Squares (STLS) (Brunton et al., 2016). For a given tolerance that filters the entries of $\boldsymbol{\Xi}$ with small value, we can obtain a sparse representation of $\mathcal{F}$. Iterative search with STRidge can be performed to find the optimal tolerance according to the selection criteria:

$$\boldsymbol{\Xi}^* = \underset{\boldsymbol{\Xi}}{\arg\min}\left\{||\mathbf{U}_t - \boldsymbol{\Theta}(\mathbf{U})\boldsymbol{\Xi}||_2 + \gamma||\boldsymbol{\Xi}||_0\right\} \tag{4}$$

where $||\boldsymbol{\Xi}||_0$ is used to measure the sparsity of coefficient vector; $||\mathbf{U}_t - \boldsymbol{\Theta}(\mathbf{U})\boldsymbol{\Xi}||_2$ denotes the regression error; $\gamma$ is the coefficient that balances the sparsity and the regression errors. Since the optimization objective has two components, we apply Pareto analysis to select an appropriate $\gamma$ (see Appendix Section H). This procedure to find a sparse $\boldsymbol{\Xi}$ is also known as sparse regression.

### 3.4 HYBRID PEDL FRAMEWORK FOR EQUATION DISCOVERY

In this part, we present the hybrid/coupled scheme of the proposed PeRCNN and the sparse regression for discovering the governing PDEs. As shown in Fig. 1, the whole equation discovery process can be divided into three stages, including the metadata reconstruction, the sparse regression for PDE discovery and the fine-tuning of PDE coefficients. The proposed framework enables us to iterate over these three stages to keep refining the result until convergence. The purpose of each stage, as well as the technical details, is detailed as follows.

**Data reconstruction.** Since the experimental measurements are typically sparse and noisy, we firstly utilize the PeRCNN introduced in Section 3.2 to reconstruct the high-fidelity data in order to obtain accurate derivative terms in the stage of sparse regression. Given the measurement[2] $\tilde{\mathbf{u}} \in \mathbb{R}^{n_t' \times H' \times W'}$ on a coarse grid (with $H' \times W'$ resolution) at $n_t'$ time steps, we establish a data-driven model that minimizes the misfit error (i.e., mean squared error) between the predictions from the PeRCNN model and the input measurements. In specific, the loss function is given by

$$\mathcal{L}(\boldsymbol{\theta}) = \text{MSE}\left(\widehat{\boldsymbol{\mathcal{U}}}(\tilde{\mathbf{x}}) - \tilde{\mathbf{u}}\right) + \eta\text{MSE}\left(\widehat{\boldsymbol{\mathcal{U}}}^{(0)} - \mathcal{I}(\tilde{\mathbf{u}}^{(0)})\right) \tag{5}$$

Note that the 1st part in RHS of Eq. (5) is adopted for training the network based on the LR data $\tilde{\mathbf{u}}$. Since we use $\tilde{\mathbf{x}}$ to denote the set of locations (i.e., coarse grid) on which the LR data is collected, $\widehat{\boldsymbol{\mathcal{U}}}(\tilde{\mathbf{x}})$ denotes the mapping of HR prediction $\widehat{\boldsymbol{\mathcal{U}}} \in \mathbb{R}^{n_t \times H \times W}$ on the coarse grid $\tilde{\mathbf{x}}$, where $n_t$ is the number of time steps for reconstructed data and $H \times W$ the resolution for the fine grid. Besides, the 2nd part in RHS of Eq. (5) is the regularizer term referring to the discrepancy of measurement-interpolated HR initial state $\mathcal{I}(\tilde{\mathbf{u}}^{(0)})$ and the predicted HR initial state $\widehat{\boldsymbol{\mathcal{U}}}^{(0)}$ from ISG, where the

---

[2]$\tilde{\mathbf{u}}$ herein denotes the LR measurement while $\mathbf{u}$ in Eq. (1) represents the symbol of state variable.

superscript "0" indicates the first snapshot of the measurement or prediction. The introduction of the initial condition (IC) regularizer is because of its effectiveness in avoiding the model overfitting. Moreover, $\eta$ is the weighting coefficient for balancing these two loss components in the model training. We first pretrain ISG to ensure the accuracy of the HR initial state (i.e., $\widehat{\mathcal{U}}^{(0)}$). Then the whole network (ISG and recurrent $\Pi$-block) is trained using the loss function of Eq. (5).

**Sparse regression.** With the reconstructed high-fidelity (i.e., HR and denoised) solution, we are able to reliably and accurately perform sparse regression for the explicit form (analytical structure) of PDEs. As described in Section 3.3, given a library of candidate functions $\boldsymbol{\Theta}(\mathbf{U})$, sparse regression seeks to find a suitable coefficient vector $\boldsymbol{\Xi}$ that balances model complexity and accuracy. This is realized by solving the optimization problem described by Eq. (4) with the STRidge algorithm. Note that sparse regression is performed on subsampled $\boldsymbol{\Theta}(\mathbf{U})$ (i.e., 10% rows are randomly sampled) to avoid a very large library matrix that would lead to extremely slow computation or exceeding computer's memory limit. Similar strategy is adopted in PDE-FIND (Rudy et al., 2017).

**Fine-tuning of coefficients.** The obtained coefficients from sparse regression may not fully exploit all the available measurement as the regression is performed on subsampled HR data. To further improve the result of equation discovery, we present a fine-tuning step to produce the final explicit governing equation. In the fine-tuning step, all the spatiotemporal measurements are used to train a recurrent block completely based on the identified PDE structure from the sparse regression (see Fig. 1(3b)). The coefficient of each term, which is treated as a trainable variable in the network, can be obtained by minimizing the misfit error between the predictions and measurements. In Appendix Section G, we show that fine-tuning can effectively improve the accuracy of the discovered PDE.

## 4 EXPERIMENTS

### 4.1 DATASETS

To examine the effectiveness of the proposed approach, we test on three different datasets (i.e., PDE systems) which cover the 2D Burgers' equation, 2D Lambda-Omega ($\lambda$–$\Omega$) and 2D Gray-Scott (GS) reaction-diffusion (RD) equations. All of them are valuable mathematically and practically, and usually work as the benchmark examples. For instance. 2D Burgers' equation has wide applications in fluid mechanics, acoustics and traffic flow, while the GS RD equation is extensively used to model the process in chemistry and biochemistry (whose response has very complex patterns). Detailed description of each equation is provided in Appendix Section B.

### 4.2 EVALUATION METRICS

We adopt three quantitative metrics to examine the performance of our method for PDE discovery:

**Relative $\ell_2$ error of coefficients.** The relative $\ell_2$ error, defined as $E = ||\boldsymbol{\Xi}_{\text{id}} - \boldsymbol{\Xi}_{\text{true}}||_2/||\boldsymbol{\Xi}_{\text{true}}||_2$, measures the relative distance between the identified coefficient vector $\boldsymbol{\Xi}_{\text{id}}$ and the ground truth $\boldsymbol{\Xi}_{\text{true}}$. However, when the magnitude of coefficients vary significantly (e.g., the GS RD equation), $E$ cannot reflect the result well as the small coefficients get overwhelmed. Hence, we introduce the following non-dimensional measures to gain a more insightful evaluation of the results.

**Precision and recall.** The PDE discovery can also be considered as a binary classification problem (e.g., whether a term exists or not) given a candidate library. Therefore, we present the precision and recall to evaluate the performance of the proposed method. The recall measures the percentage of the successfully identified coefficients among the true coefficients, defined as $R = ||\boldsymbol{\Xi}_{\text{id}} \odot \boldsymbol{\Xi}_{\text{true}}||_0/||\boldsymbol{\Xi}_{\text{true}}||_0$ where $\odot$ denotes element-wise product of two vectors[3]. Similarly, the precision has the definition of $P = ||\boldsymbol{\Xi}_{\text{id}} \odot \boldsymbol{\Xi}_{\text{true}}||_0/||\boldsymbol{\Xi}_{\text{id}}||_0$.

### 4.3 EXPERIMENT SETUP

**Generation of measurements.** The noisy and LR measurements are synthetized through numerical simulations with high-order FD methods. 9-point stencil is used for computing the spatial deriva-

---

[3]A successful identification occurs when the entries in both identified and true vectors are nonzero.

tives while time marching is performed through Runge-Kutta scheme. For the 2D Burgers' equation system, we select $\nu$ to be 0.005 (Geneva & Zabaras, 2020) and generate the solution on a unit square domain discretized by the $101 \times 101$ grid. 200 time steps are simulated for a total duration of 0.05 second. The numerical solution is downsampled to measurements as $\tilde{\mathbf{u}} \in \mathbb{R}^{40 \times 2 \times 51 \times 51}$. For the 2D $\lambda$–$\Omega$ RD system, $\beta$, $\mu_u$ and $\mu_v$ are set to be 1, 0.1 and 0.1 respectively. We discretized the $20 \times 20$ square domain into the $101 \times 101$ grid. 200 time steps are considered for a physical duration of 2.5 seconds. The measurements are downsampled from the solution to be $\tilde{\mathbf{u}} \in \mathbb{R}^{40 \times 2 \times 51 \times 51}$. For the 2D GS RD system, the solution is generated on the $101 \times 101$ grid for a unit square domain using the parameters of $\mu_u = 2 \times 10^{-5}$, $\mu_v = 5 \times 10^{-6}$, $\kappa = 0.06$ and $f = 0.04$. 800 time steps are simulated for the total duration of 400 seconds. The measurements are obtained by downsampling the numerical solution to $\tilde{\mathbf{u}} \in \mathbb{R}^{160 \times 2 \times 26 \times 26}$. Periodic boundary condition is adopted for all three systems. In addition, to mimic the measurement noise in the real world, we add the Gaussian noise of a given level (e.g., 5% and 10%) to the numerical solution before downsampling for the LR data.

**Baselines.** In the experiments, we compare the performance of our proposed method with the widely used PDE-FIND (Rudy et al., 2017). Different from the settings in (Rudy et al., 2017), we assume only LR measurements are available for computing the derivatives in $\boldsymbol{\Theta}(\mathbf{U})$. Specifically, the FD is used to obtain the derivatives for noiseless data. However, in the case that data is heavily corrupted by noise, the derivatives of the fitted Chebyshev polynomial are used. In addition, we also consider the baselines of the sparse regression coupled with a fully connected neural network (namely FCNN+SR) or PDE-Net[4] (namely PDE-Net+SR). In these two baselines, the FCNN and PDE-Net (Long et al., 2018) are used respectively to fit the LR measurement and perform inference for HR data while SR is used to discover the PDE. Notably, automatic differentiation (Baydin et al., 2017) is used in FCNN+SR to compute the partial derivatives in $\boldsymbol{\Theta}(\mathbf{U})$. However, like many other methods (Chen et al., 2021; Xu et al., 2021) that utilize the FCNN to fit the sparse data, FCNN+SR lacks the capability of encoding prior physics knowledge. Therefore, the prior knowledge of known diffusion terms would not be utilized in the data reconstruction phase of FCNN+SR.

## 4.4 RESULTS

**2D Burgers' equation.** We first test the proposed approach on the 2D Burgers' equation. To reconstruct the high-fidelity data, we use the PeRCNN with 3 parallel Conv layers of 16 channels and filter size of 5. These hyperparameters of network architecture are selected through hold-out validation. The range for selecting the network hyperparameters are given in Appendix Section C. Furthermore, in this and the following examples, we assume the measurement exhibits the ubiquitous diffusion phenomenon. That said, the diffusion layer would be encoded into the network architecture. Also in the sparse regression stage, the coefficient of diffusion term would be exempted from being filtered. We train the network with Adam optimizer for 15,000 iterations. The learning rate is initialized to be 0.002 and decreases to 97% of the previous for every 200 iterations.

Once the training is completed, the HR solution (i.e., 201 snapshots of $101 \times 101$) can be inferred from the trained model. Figure 2(a) provides the snapshots of the reconstructed HR data under 5% noise from each method. It can be seen that our method and the FCNN has a much smaller reconstruction error, which would give rise to more accurate derivatives for constructing the library matrix. We also observe that PDE-Net struggles in reconstructing the HR data. Based on our experiments and the claim PDE-Net paper (Long et al., 2018) made, this is because the additive operation limits PDE-Net to only learning the response of linear PDEs. Therefore, unlike our model, PDE-Net lacks the capability to exactly express nonlinear terms like $uu_x$ and $u^2v$. To prepare for the subsequent sparse regression, we established a group of 70 candidate functions that consists of polynomial terms $\{1, u, v, u^2, uv, v^2, u^3, u^2v, uv^2, v^3\}$, derivatives $\{1, u_x, u_y, v_x, v_y, \Delta u, \Delta v\}$ and their combinations. The selection of the crucial hyperparameter $\gamma$ in sparse regression is conducted through Pareto analysis, which is detailed in Appendix Section H. In addition to the diffusion terms known *a priori*, the result of sparse regression indicates the presence of several other terms in $\mathcal{F}$, i.e., $\mathcal{S}_u = \{uu_x, vu_y\}$ and $\mathcal{S}_v = \{uv_x, vv_y\}$ respectively. As the last step, the fine-tuning is performed using the network built completed based on the discovered PDE structure in the second

---

[4]As the original PDE-Net does not account for LR measurement data, we replace the $\Pi$-block in our model with the $\delta t$-block in PDE-Net to make it a HR predictive model.

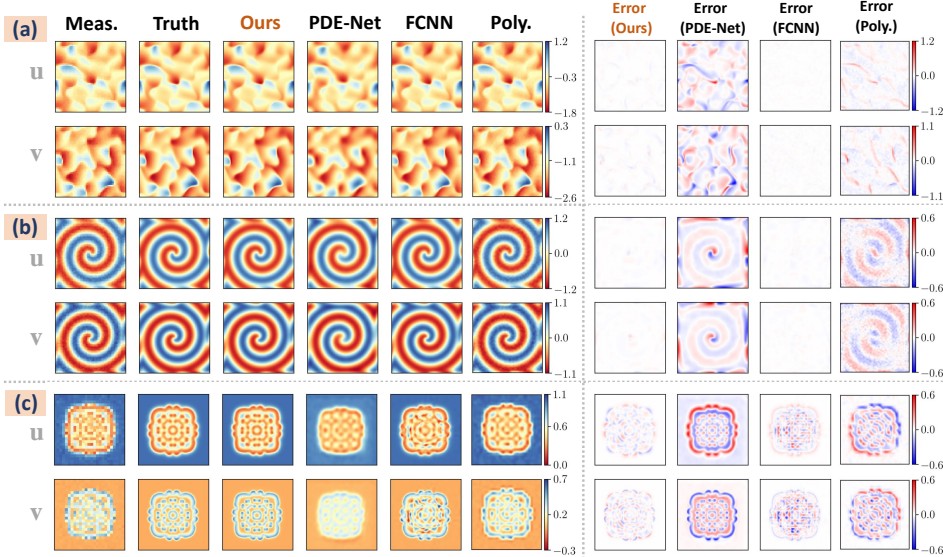

Figure 2: Snapshots of the state variable at one time instance. (a)-(c) represent the 2D Burgers', $\lambda$–$\Omega$ and GS RD systems. Six columns starting from the left denote the LR measurement, HR ground truth, reconstructed HR solution from our model, PDE-Net, FCNN and Chebyshev polynomial fitting (i.e., PDE-FIND) respectively. The reconstruction error of each model is also provided.

stage (see Fig. 1(3b)), which gives us the final discovered equation:

$$u_t = 5.0113 \times 10^{-3} \Delta u - 1.004 u u_x - 1.004 v u_y,$$
$$v_t = 4.9953 \times 10^{-3} \Delta v - 1.009 u v_x - 1.002 v v_y. \tag{6}$$

The precision, recall and relative $\ell_2$ error of the discovered PDE under various noise levels are provided in Table 1. It can be observed that all methods except PDE-Net+SR perform well when the measurement is noise-free. Furthermore, the performance of baselines deteriorate when the noise level increases to 10% while our approach demonstrates better robustness against the noise. Our approach achieves $5.9 \times 10^{-3}$ relative $\ell_2$ error, 100% recall and precision under the 10% noise level. In the Appendix Sections E and F, we demonstrate that our method could handle even larger noise (up to 30%) and sparser data. We also consider a more challenging case (increasing the Reynolds number to 500 for the Burgers' equation), in which our method still work well for both response reconstruction and PDE discovery (see Appendix Section I). We also show the interpretability of the trained $\Pi$-block network for the 2D Burgers' case (see Appendix Section J).

**2D $\lambda$–$\Omega$ RD equation.** In this example, the recurrent network employed for data reconstruction has 3 parallel Conv layers, 16 channels and $5 \times 5$ kernels. The same procedure in previous example for training the reconstruction network is adopted here. Once the training is completed, we infer from the network the HR solution at finer spatiotemporal grid. The reconstruction error of our method and the PDE-FIND is shown in Fig. 2(b). The same candidate set in 2D Burgers' example (with 70 candidate functions) is adopted to establish the library matrix $\Theta(\mathbf{U})$. We randomly subsample 10% of the HR points for the sparse regression, whose result (5% noise case) gives the set of terms with nonzero coefficients, i.e., $\mathcal{S}_u = \{\Delta u, u, u^3, u^2 v, uv^2, v^3\}$ and $\mathcal{S}_v = \{\Delta v, u, u^3, u^2 v, uv^2, v^3\}$. With the set of existing terms in the governing PDE, the fine-tuning step is performed using all the available measurement. The obtained scalar coefficients from the fine-tuning renders us with the final governing equation, which reads as

$$u_t = 0.096 \Delta u + 1.038 u - 1.048 u^3 + 1.004 u^2 v - 1.050 uv^2 + 0.998 v^3,$$
$$v_t = 0.100 \Delta v + 1.014 v - 0.998 u^3 - 1.025 u^2 v - 0.999 uv^2 - 1.015 v^3. \tag{7}$$

The evaluation metrics computed from the discovered PDE under various noise levels are provided in Table 1. Our approach achieves $5.44 \times 10^{-2}$ relative error, 100% recall and 91.6% precision under 10% noise. Our method is also capable of handling much sparser data (Appendix Section F).

Table 1: Performance comparison between our proposed framework and baselines.

| Cases | Metrics | Relative $\ell_2$ error $/\times 10^{-2}$ | | | Precision /% | | | Recall /% | | |
|---|---|---|---|---|---|---|---|---|---|---|
| | Noise level | 0% | 5% | 10% | 0% | 5% | 10% | 0% | 5% | 10% |
| Burgers' | Ours | 0.50 | 0.54 | **0.59** | **100** | **100** | **100** | **100** | **100** | **100** |
| | FCNN+SR | **0.37** | **0.43** | 1.34 | **100** | **100** | 75.0 | **100** | **100** | **100** |
| | PDE-Net+SR | 339.7 | 442.3 | 372.0 | 40.0 | 25.0 | 28.5 | 66.7 | 50.0 | 33.3 |
| | PDE-FIND | 3.32 | 36.80 | 45.09 | 75 | **100** | 62.5 | **100** | 83.3 | 83.3 |
| $\lambda$–$\Omega$ RD | Ours | **1.18** | **2.69** | **5.44** | **100** | **100** | **91.6** | **100** | **100** | **100** |
| | FCNN+SR | 5.19 | 7.50 | 15.85 | **100** | 85.7 | 85.7 | **100** | **100** | **100** |
| | PDE-Net+SR | 62.23 | 107.45 | 104.63 | 81.82 | 64.3 | 50.0 | 75.0 | 75.0 | 50.0 |
| | PDE-FIND | 1.70 | 92.52 | 99.15 | **100** | 83.3 | 77.8 | **100** | 62.5 | 58.3 |
| GS RD | Ours | **1.59** | **2.85** | **10.03** | **100** | **100** | **85.7** | **100** | **100** | **85.7** |
| | FCNN+SR | 95.14 | 143.55 | 162.98 | 37.5 | 33.3 | 33.3 | 60.0 | 57.1 | 57.1 |
| | PDE-Net+SR | 204.61 | 100.00 | 382.20 | 30.8 | 40.0 | 30.0 | 57.1 | 28.6 | 42.9 |
| | PDE-FIND | 113.05 | 89.99 | 108.1 | 45.5 | 50.0 | 42.9 | 85.7 | 57.1 | 60.0 |

**2D GS RD equation.** In the last example, we assume the measurements are some extremely LR (i.e., $26 \times 26$) snapshots (see measurements in Fig. 2(c)). To remedy the lack of resolution in spatial dimension, we further introduce the assumption that reaction term is in the form of polynomial after reviewing the existing literatures. As the first step, we employ the network with 3 parallel Conv layers, 8 channels and filter size of 1 to reconstruct the data. From the reconstruction model, we infer the HR (i.e., $101 \times 101$) solution at finer time instances. Snapshots of HR solution in Fig. 2(c) show our method is able to restore the high-fidelity data from the LR and noisy measurement very well. With the HR solution, a library of polynomials up to the third degree $\Theta(u, v) = [1, u, v, u^2, uv, v^2, u^3, u^2v, uv^2, v^3]$ is constructed for sparse discovery. We subsample 10% HR data for the sparse regression, which gives us the remaining terms $\mathcal{S}_u = \{\Delta u, 1, u, uv^2\}$ and $\mathcal{S}_v = \{\Delta v, v, uv^2\}$. With the existing terms in the PDE, we perform the fine-tuning using the completely physics-based Conv block. The final obtained PDEs in the case of 5% noise are

$$
\begin{aligned}
u_t &= 2.001 \times 10^{-5} \Delta u - 1.003 uv^2 - 0.04008 u + 0.04008, \\
v_t &= 5.042 \times 10^{-6} \Delta v + 1.009 uv^2 - 0.1007 v.
\end{aligned}
\tag{8}
$$

As presented in Table 1, our proposed approach performs well on discovering the PDE as a result of fully utilizing the prior physics knowledge and the powerful expressiveness of the model, while PDE-FIND , PDE-Net and FCNN+SR struggle due to the extremely LR measurements.

## 5 CONCLUSION

We propose a hybrid computational framework for discovering nonlinear PDE systems from sparse (LR) and noisy measurement data, which couples the novel network architecture PeRCNN for the data-driven modeling of dynamical systems and the sparse regression for distilling the dominant candidate functions and coefficients. First of all, one major innovation of our framework to scientific discovery comes from the nonlinearity achieved by the element-wise product operation and its powerful expressiveness (see Appendix Section A) for the nonlinear function $\mathcal{F}(\cdot)$ in PDEs. Besides, our network also features the capability to encode prior physics knowledge via a highway Conv layer that helps to overcome the challenges brought by the scarcity and noise of the measurements. Secondly, we successfully marry PeRCNN to the sparse regression algorithm to solve the crucial equation discovery issues. The coupled scheme enables us to iteratively optimize the network parameters, and fine-tune the discovered PDE structures and coefficients. Through the numerical validation, we demonstrate the superior reconstruction capability and excellent discovery accuracy of our proposed framework against three baseline approaches. Overall, we provide an effective, interpretable and flexible approach to accurately and reliably discover the underlying physical laws from the imperfect and coarse-meshed measurements.

ACKNOWLEDGEMENT

The work is supported in part by the Beijing Outstanding Young Scientist Program (No. BJJWZYJH012019100020098) as well as the Intelligent Social Governance Platform, Major Innovation & Planning Interdisciplinary Platform for the "Double-First Class" Initiative, Renmin University of China.

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

APPENDIX

## A  Π-BLOCK AS A UNIVERSAL POLYNOMIAL APPROXIMATOR

In the proposed PeRCNN, Π-block acts as an universal polynomial approximator to unknown non-linear functions while the physics-based highway Conv layer (i.e., with FD stencil as the Conv filter) accounts for the prior knowledge on the governing equation. Notably, the Π-block achieves its non-linearity through the element-wise product operation (see Eq. (2)) instead of what is widely adopted by the traditional Conv network, in which nonlinear layers are sequentially interwined with linear layers. Compared with the additive form representation $\mathcal{F}(\mathbf{u}) = \sum_{0 \le i+j \le N} f_{ij} \cdot (\mathcal{K}_{ij} \circledast \mathbf{u})$ seen in related work (Long et al., 2018; Guen & Thome, 2020), this multiplicative representation of Π-block promotes the network expressiveness for nonlinear functions. To support this claim, we need to recognize that Π-block roots on the numerical differentiation (i.e., forward Euler time stepping), which can guarantee the convergence of the solution under two conditions: (i) the RHS $\mathcal{F}$ of Eq. (1) can be computed accurately; and (ii) the time spacing $\delta t$ is sufficiently small. As the second condition can be satisfied by selecting an appropriate $\delta t$, we would show that, by construction, our Π-block can approximate any nonlinear polynomial involving spatial derivatives, such as $uu_x + vu_y$ and $u\Delta u$. This can be easily argued by proving that each monomial (e.g., $uu_x$, $vu_y$, etc.) can be approximated by multiplying the output of given number of Conv layers. To this end, we provided the following lemma and its proof reproduced from (Long et al., 2019).

***Lemma 1:*** By construction, a convolutional filter $\mathcal{K}$ with a given size can approximate any differential operator with prescribed order of accuracy.

***Proof 1:*** Without loss of generality, let us consider a bivariate differential operator $\mathcal{L}(\cdot)$ w.r.t variables $x$ and $y$, we have

$$
\begin{aligned}
\mathcal{L}(u) &= \sum_{k_1,k_2=-\frac{N-1}{2}}^{\frac{N-1}{2}} \mathcal{K}[k_1,k_2] \sum_{i,j=0}^{N-1} \left.\frac{\partial^{i+j} u}{\partial^i x \partial^j y}\right|_{(x,y)} \frac{k_1^i k_2^j}{i!j!} \delta x^i \delta y^j + \mathcal{O}(|\delta x|^{N-1} + |\delta y|^{N-1}) \\
&= \sum_{k_1,k_2=-\frac{N-1}{2}}^{\frac{N-1}{2}} \mathcal{K}[k_1,k_2] u(x+k_1\delta x, y+k_2\delta y) + \mathcal{O}(|\delta x|^{N-1} + |\delta y|^{N-1}) \\
&= \mathcal{K} \circledast u + \mathcal{O}(|\delta x|^{N-1} + |\delta y|^{N-1})
\end{aligned}
\tag{9}
$$

where $N$ is the size of the filter. The weight matrix $\mathcal{K}[\cdot, \cdot]$ is indexed by $k_1$ and $k_2$. Letting the filter's entry $\mathcal{K}[k_1, k_2]$ be the corresponding Taylor series coefficient, we can see the approximation error is bounded by $\mathcal{O}(|\delta x|^{N-1} + |\delta y|^{N-1})$.

Therefore, compared with the traditional black-box models (e.g., deep neural networks) for representing nonlinear functions, Π-block possesses two main advantages:

- The nonlinear function $\mathcal{F}$ in the form of multivariate polynomial (e.g., $\mathbf{u} \cdot \nabla u + u^2 v$) covers a wide range of well-known dynamical systems, such as Navier-Stokes, reaction-diffusion (RD), Lorenz equations, to name only a few. Since the spatial derivatives can be computed by Conv kernels, a Π-block with $n$ parallel Conv layers of appropriate filter size is able to represent a polynomial up to the $n^{\text{th}}$ order.
- The Π-block is flexible in representing the nonlinear function $\mathcal{F}$. For example, a Π-block with 2 parallel layers of appropriate filter size ensembles a family of polynomials up to the $2^{\text{nd}}$ order (e.g., $u$, $\Delta u$, $uv$, $\mathbf{u} \cdot \nabla u$), with no need to explicitly define the basis functions.

## B  DETAILED DESCRIPTION OF THE CONSIDERED PDEs

The first 2D Burgers' equation studied in this paper has the form of
$$
\mathbf{u}_t + \mathbf{u} \cdot \nabla \mathbf{u} = \nu \Delta \mathbf{u}
\tag{10}
$$

where $\nabla$ is the Nabla operator; $\Delta$ is the Laplacian operator; $\mathbf{u} = [u, \ v]^{\mathrm{T}}$ is the fluid velocities and $\nu$ is the viscosity coefficient. The second and third RD equation system can both be described by

$$\mathbf{u}_t = \mathbf{D}\Delta\mathbf{u} + \mathbf{R}(\mathbf{u}) \tag{11}$$

where $\mathbf{u} = [u, \ v]^{\mathrm{T}}$ is the concentration vector; $\mathbf{D} = [\mu_u, 0; 0, \mu_v]$ is the diagonal diffusion coefficient matrix; $\mathbf{R}(\mathbf{u})$ denotes the reaction vector. For the $\lambda$–$\Omega$ RD system, the reaction vector has the form of $\mathbf{R}(\mathbf{u}) = [(1 - u^2 - v^2)u + \beta(u^2 + v^2)v, \ -\beta(u^2 + v^2)u + (1 - u^2 - v^2)v]^{\mathrm{T}}$ where $\beta$ is a reaction parameter. For the GS RD system, the nonlinear reaction vector reads $\mathbf{R}(\mathbf{u}) = [-uv^2 + f(1 - u), \ uv^2 - (f + \kappa)v]^{\mathrm{T}}$ where $\kappa$ and $f$ denote the kill and feed rate respectively. The dataset and training script for each case considered in this paper can be found in https://github.com/Raocp/Discover-PDE-with-Noisy-Scarce-Data.

## C  HYPERPARAMETER SELECTION OF PERCNN IN DATA RECONSTRUCTION

The selection of the hyperparameters of PeRCNN for data reconstruction is performed through the hold-out validation. Among the entire measurement, 10% data is split as the validation dataset. The range of hyperparameters we considered are summarized in Table C.1. The value in the parenthesis denotes the final parameter adopted in the data reconstruction.

Table C.1: Range of hyperparameters for the data reconstruction network.

| Dataset | Kernel size | # layers | # channels | Learning rate | Regularizer weight $\eta$ |
|---|---|---|---|---|---|
| 2D Burgers' | 1∼5 (5) | 2∼4 (3) | 4∼16 (16) | 0.001∼0.01 (0.002) | 0.001∼1 (1) |
| 2D $\lambda$–$\Omega$ | 1∼5 (5) | 2∼4 (3) | 4∼16 (16) | 0.001∼0.01 (0.002) | 0.001∼1 (0.01) |
| 2D GS | 1∼5 (1) | 2∼4 (3) | 4∼16 (8) | 0.001∼0.01 (0.001) | 0.001∼1 (0.005) |

## D  ENCODE PHYSICS KNOWLEDGE TO ENHANCE THE MODEL

As introduced in Section 3.2, one salient characteristic of the proposed network is the capability to encode prior physics knowledge. To examine how the encoded physics enhances the accuracy of the data-driven model, here we use the 2D Burgers' equation system as a testbed. In the experiment, we consider two different cases: (1) no prior knowledge is available, and (2) the diffusion term ($\Delta\mathbf{u}$) is known to exist in $\mathcal{F}$. The setting in the second case leads to an additional physics-based Conv layer in PeRCNN to account for the diffusion term. Apart from this, other hyperparameters are kept same in the experiment.

Figure D.1 shows the learning curves of these two data-driven models. It can be observed that the network encoded with the diffusion term exhibits faster convergence and better accuracy compared with the competitor. As a result of encoding prior physics knowledge, the PeRCNN does not need to learn a diffusion operator through the back propagation of error. That said, incorporating available physics can facilitate the learning efficiency and accuracy of the data-driven model.

## E  TOLERANCE TO LARGER NOISE LEVEL

To examine the scalability of our method with regard to the Gaussian noise level, we performed additional experiments with the noise level of 20% and 30% for all three cases. The same settings from Section 4 is adopted in these experiments. The performance of our method on various noise levels is summarized in Table E.1. We observed that larger noise affects the accuracy of our method differently. For the 2D GS RD equation, the accuracy of our method deteriorates drastically as the noise level increases. That is because the measurement data used in this example has a relatively low resolution (i.e., $26 \times 26$). Even worse, the phenomenon in 2D GS RD system is characterized with local patterns and sharp gradients (see Fig. 2c). Increasing the Gaussian noise would further blur the spatial information in the measurement data. However, for the case of 2D $\lambda$-$\Omega$ RD and Burgers' equation where the measurement data has a decent resolution (i.e., $51 \times 51$) and smoother spatial patterns, our method stays competitive even for 30% Gaussian noise. Overall, there is a trade-off between the sparsity and noise level of the measurement data.

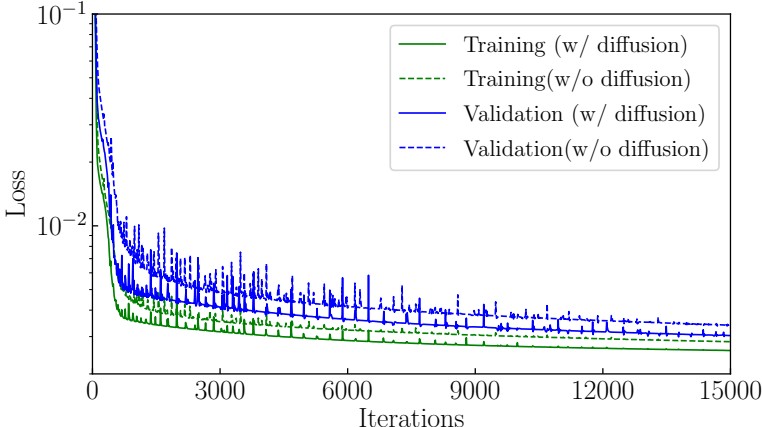

Figure D.1: Learning curve of the data-driven model with (w/) and without (w/o) encoded diffusion layer for 15,000 iterations.

Table E.1: Performance of the proposed method on larger noise level.

| Cases | Noise level /% | Precision /% | Recall /% | Relative $\ell_2$ error /$\times 10^{-2}$ |
|---|---|---|---|---|
| | 0 | 100 | 100 | 0.50 |
| | 5 | 100 | 100 | 0.54 |
| Burgers' | 10 | 100 | 100 | 0.59 |
| | 20 | 75.0 | 100 | 7.33 |
| | 30 | 75.0 | 100 | 9.10 |
| | 0 | 100 | 100 | 1.18 |
| | 5 | 100 | 100 | 2.69 |
| $\lambda$–$\Omega$ RD | 10 | 91.6 | 100 | 5.44 |
| | 20 | 91.6 | 100 | 7.65 |
| | 30 | 91.6 | 100 | 15.89 |
| | 0 | 100 | 100 | 1.59 |
| | 5 | 100 | 100 | 2.85 |
| GS RD | 10 | 85.7 | 85.7 | 10.03 |
| | 20 | 46.2 | 85.7 | 86.05 |
| | 30 | 33.3 | 37.5 | 238.31 |

## F    MEASUREMENT DATA SPARSITY

Our method assumes the measurement data is collected on a fixed coarse sensor grid (e.g., $21 \times 21$ sensors), captured either by point-wise sensor units or by imaging techniques. Although the ground truth data is of high fidelity, our measurement data is heavily spatiotemporally-downsampled and, as a result, remains sparse (e.g., a finite number of LR snapshots). As for the scalability with data sparsity, we performed numerical experiments on the 2D GS and $\lambda$–$\Omega$ RD equations using various spatial resolutions, namely, $51 \times 51$, $26 \times 26$, $21 \times 21$ and $11 \times 11$. The observation is that the tolerable sparsity of our method depends on the spatial pattern of the system. For example, in the 2D GS RD example, the lowest resolution of data to guarantee the discoverability of the governing PDE is $26 \times 26$ since the very complex maze-like pattern (see Fig. 2c) is in a relatively fine scale. However, since the solution in the $\lambda$–$\Omega$ RD example is much smoother and periodic (see Fig. 2b), our method is able to discover a major portion of the PDE (i.e., precision 90.9%, recall 83.3%) with the spatial resolution as low as $11 \times 11$ (a very sparse data scenario).

## G    IMPORTANCE OF THE COEFFICIENT FINE TUNING

Our proposed approach for discovering PDEs consists of three stages including the data reconstruction, the sparse regression and the fine-tuning of coefficients. In fact, the sparse regression has

already been able to determine the explicit form of the discovered PDE. Then a natural question would be why the last fine-tuning step is necessary. As its name indicates, the fine-tuning step is able to further improve the accuracy of the discovered PDE, for mainly two reasons: (i) The sparse regression step gives an explicit PDE based on the primitive assumption of a sparse $\Xi$. However, the discovered PDE can be further considered as the latest assumption, based on which a completely physics-based recurrent network can be established. In the fine-tuning step, the functional form of the PDE is unchanged while its scalar coefficients are optimized. That said, the fine-tuning step is an optimization process on a much smaller parameter space. (ii) The sparse regression is performed on subsampled high-dimensional spatiotemporal data while the fine-tuning step could utilize all the available measurement through the recurrent network.

The above argument provides some intuitions of the coefficient fine-tuning step. To further justify it, we compute the relative $\ell_2$ error of the coefficient vector before/after the fine-tuning step, which is given in Table G.1. It can be seen that fine-tuning step improves the accuracy of the discovered PDE consistently for different noise level. Interestingly, we observe that for the 2D GS and $\lambda$–$\Omega$ RD case under 10% noise, the improvement brought by the fine tuning step is not as evident as that of the other cases. That is because, in these two cases, the sparse regression does not give the correct form of the PDE. In other words, the fine-tuning is performed based on an assumption that deviates from the fact. The dependency on the sparse regression result also implies one limitation of our approach.

Table G.1: Relative $\ell_2$ error (unit: $10^{-2}$) of the coefficient vector before and after the fine tuning.

| Case | Stage | Noise level | | |
|---|---|---|---|---|
| | | 0% | 5% | 10% |
| Burgers' | Before | 0.93 | 1.57 | 2.18 |
| | After | 0.50 | 0.54 | 0.59 |
| $\lambda$–$\Omega$ RD | Before | 1.52 | 3.33 | 6.65 |
| | After | 1.18 | 2.69 | 5.44 |
| GS RD | Before | 6.40 | 9.84 | 12.33 |
| | After | 1.59 | 2.85 | 10.03 |

## H    PARETO ANALYSIS FOR HYPERPARAMETER SELECTION

As introduced in Section 3.3, sparse regression is formulated as an optimization problem described by Eq. (4). The aggregated optimization objective consists of two components, $||\mathbf{U}_t - \boldsymbol{\Theta}(\mathbf{U})\boldsymbol{\Xi}||_2$ and $||\boldsymbol{\Xi}||_0$, to ensure the accuracy of the regression while keeping the model parsimonious. This trade-off can be balanced by selecting an appropriate weighting coefficient $\gamma$ in the aggregated objective function. Pareto analysis is so far considered as the most effective way for selecting this hyperparameter $\gamma$. To ensure the scale of two components aligned, we introduce an equivalent hyperparameter $\kappa$ as the nondimensional weighting coefficient. The $\kappa$ and $\gamma$ are interconvertible with the definition $\gamma \equiv \kappa||\mathbf{U}_t - \boldsymbol{\Theta}(\mathbf{U})\tilde{\boldsymbol{\Xi}}||_2$ where $\tilde{\boldsymbol{\Xi}}$ is the Least Squares solution to the linear system.

In the Pareto analysis, we would sample some points of $\kappa$ from $[10^{-2}, 20]$. For each of the point, we would pass it as the argument for the sparse regression routine, which would return the optimal coefficient vector $\boldsymbol{\Xi}^*$. The corresponding regression error and $\ell_0$ penalty can be computed for Pareto analysis. Figure H.1 provides the results of Pareto analysis for 2D $\lambda$–$\Omega$ equation system (10% noise case). From the subfigure on the left, we can identify the Pareto front that corresponds to the optimal region of $\kappa$. Pareto analyses as such are performed on each case to find the suitable weighting coefficient $\kappa$ (or equivalently $\gamma$). The value of $\kappa$ employed in each case is summarized in Table H.1.

However, readers should note that the Pareto front may not always be so obvious to identify. In many scenarios, trade-off between the model sparsity and accuracy must be done with manual selection, which still appears to be more art than science.

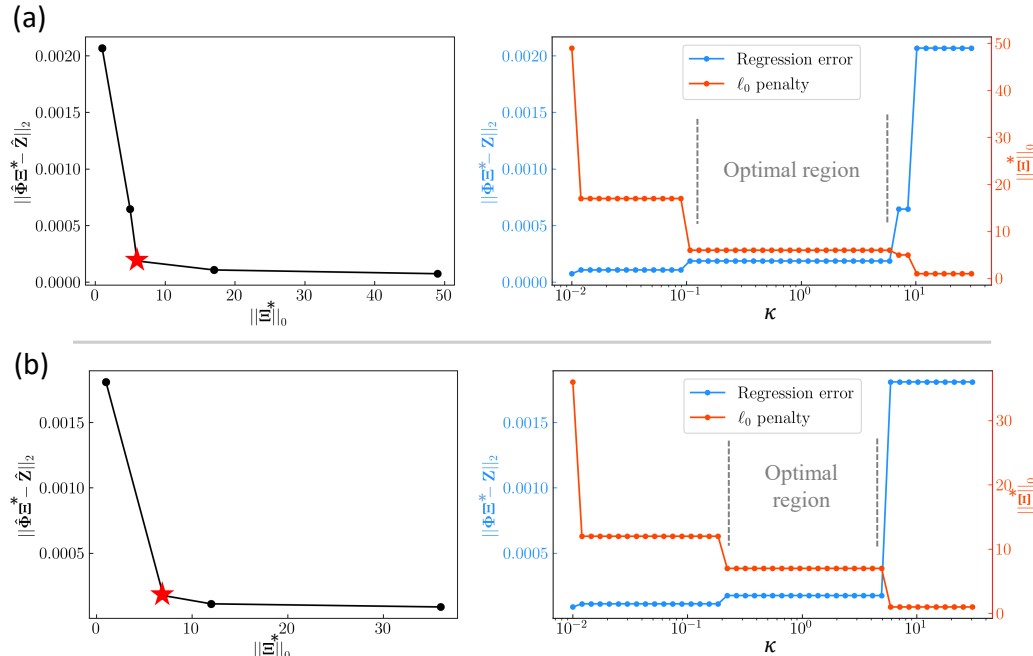

Figure H.1: Pareto analysis for selecting the weighting coefficient $\kappa$ (or $\gamma$) in 2D $\lambda$–$\Omega$ GS system (10% noise level). (a) and (b) correspond to the $\mathcal{F}_u$ and $\mathcal{F}_v$ respectively. The Pareto front that represents the optimal $\kappa$ is marked with the red star.

Table H.1: The coefficient $\kappa$ employed in sparse regression.

| Method | Burgers' | | | $\lambda$–$\Omega$ RD | | | GS RD | | |
|---|---|---|---|---|---|---|---|---|---|
| | 0% | 5% | 10% | 0% | 5% | 10% | 0% | 5% | 10% |
| Ours | 1 | 1 | 1 | 1 | 1 | 1 | 0.1 | 0.1 | 1 |
| PDE-FIND | 1 | 1 | 1 | 1 | 1 | 1 | 0.1 | 0.05 | 0.1 |

## I   Discovery of Burgers' equation under higher Reynolds number

In this part, we examine the performance of our method on Burgers' equation when Reynolds number grows. Higher Reynolds number would complicate the equation discovery problem due to the intense convective phenomena (or even chaos). We increase the Reynolds number of the problem to 500 by setting the $\nu = 0.002$ as both the average initial velocity magnitude and characteristic length is 1. 10% Gaussian noise is adopted in this example while the remaining setting follows the Section 4.4. One snapshot of the reconstructed HR data, altogether with the ground truth, is presented in Fig. I.1. It can be observed that the reconstructed HR agree well with the ground truth. Then the reconstructed HR data is used in sparse regression to obtain the structure of the governing PDE, based on which the PeRCNN is built for coefficient fine tuning. The discovered PDEs after the coefficient fine tuning are

$$u_t = 1.9854 \times 10^{-3}\Delta u - 1.0072 u u_x - 1.006 v u_y,$$
$$v_t = 1.9409 \times 10^{-3}\Delta v - 0.9953 u v_x - 1.008 v v_y. \tag{12}$$

which match the ground truth correctly.

## J   Interpret the learned model

One major drawback of the traditional deep neural network is the lack of interpretability as the network's output is usually expressed as a prolonged nested function. However, since each channel of

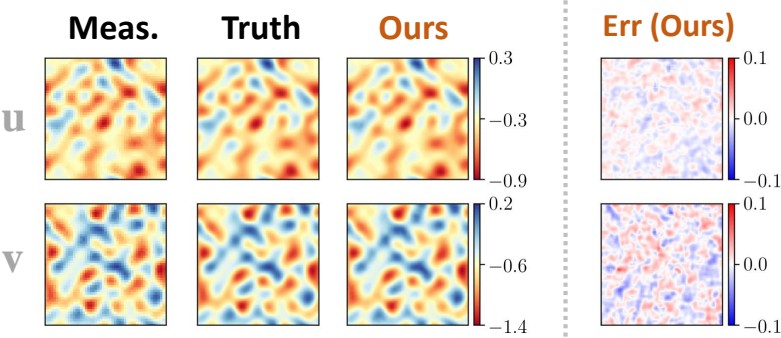

Figure I.1: Snapshots of the state variable components (i.e., $u$ and $v$) for Burgers' equation system with Reynolds number equals to 500.

the input to $\Pi$-block (i.e., $\widehat{\mathcal{U}}^{(k)}$) corresponds to a state variable component (i.e., $[u, v]$), the multiplicative form of $\Pi$-block makes it possible to extract (or interpret) the explicit form of learned $\mathcal{F}$ from the learned weights and biases via symbolic computations. This is a notable advantage of the proposed $\Pi$-block as an analytical expression would help researcher to disentangle the underlying physics or make inference on different cases (e.g., different I/BCs).

Here a simple experiment is conducted on the 2D Burgers' case. The network employed in the experiment has two Conv layers with two channels. Two channels of the first Conv layer are associated with $\partial/\partial x$ and $\partial/\partial y$ respectively, by fixing the filters with the corresponding FD stencils. The remaining settings are kept the same as the 2D Burgers' case in Section 4.4. The interpreted expression from the recurrent block (i.e., $\Pi$-block and the highway diffusion layer) model is

$$\mathbf{u}_t = \begin{bmatrix} 0.0051\Delta u - 0.95u_x(1.07u - 0.0065v - 0.17) + 0.98u_y(0.0045u - 1.01v + 0.17) + 0.053 \\ 0.0051\Delta v - 0.82v_x(1.22u + 0.0078v - 0.18) - 0.91v_y(0.0063u + 1.08v - 0.17) + 0.058 \end{bmatrix} \quad (13)$$

It is seen that the equivalent expression of the learned model is close to the genuine governing PDEs, which helps to explain the extraordinary expressiveness of our model. Although the selection of differential operators to be frozen is crucial for identifying the genuine form of the $\mathcal{F}$, this example demonstrates the better interpretability of PeRCNN over "black-box" models. Similarly, with the trained model from Section 4.4 for 2D GS RD equation which has three Conv layers and $1 \times 1$ filters, we could extract a third degree polynomial for the reaction term in $\mathcal{F}$.

