# OpenReview forum: "Discovering Nonlinear PDEs from Scarce Data with Physics-encoded Learning"
_ICLR.cc/2022/Conference — ICLR 2022 Poster_

### Official Review · Reviewer_zdB8 · 2021-10-30

**Correctness:** 2
**Technical Novelty And Significance:** 2
**Empirical Novelty And Significance:** 2
**Recommendation:** 5
**Confidence:** 5

**Main Review:**

### Updates after discussions
My concerns are partly resolved and I raise my score to 5. Detailed can be found in the discussion threads below.

Generally this paper is clearly-written and the PDE discovery problem is of great importance in the pioeering research fields. This paper uses some deep cov-net and sparse learning techniques to learn the promising results.

### Main concerns
The main concern of mine is the novelty and experiments.

* PeRCNN is proposed in the paper to generate HR data from LR data. However, this method is similar to the existing idea of directly building a neural network to generate meta data, such as DL-PDE[1]. So how much does a more complex network model (i.e., PeRCNN) improve the performance? In the experiment corresponding to Figure 2 and Table 1, the authors should consider comparing the performance of PeRCNN with the method of using a simple fully connected neural network to generate HR data. In fact, many algorithms based on sparse regression or genetic algorithms are already optimized for LR data in PDE discovery.

* The authors are suggested to compare more existing studies to show the advantage of the proposed method. Although PDE-FIND is classic and effective, it is not robust enough to noise and requires that the candidate set contain the correct solution. Therefore, there have been many new attempts in recent years, and the author should compare at least some of these studies. For example, the DLGA that also uses NN to reduce the interference of noise on the model [1], the EPDE that uses genetic algorithm to expand the candidate set [2], the AI Feynman that based on symbolic regression [3], and the SGA that can mine open-form equations form data without prior knowledge of the function terms [4], and the KO-PDE that can discover highly nonlinear coefficients of the PDEs [5], etc. Besides, authors could discuss the difference between the Π-block and PDEnet/PDEnet2.0 [6, 7] in the data reconstruction process.


* When faced with noisy data, there are two common ways to calculate the gradient/function term: i). Use neural networks to generate meta data, similar to the PeRCNN, and then use the difference method to calculate the gradients.ii). Directly use the automatic differentiation process of the neural network to generate the gradients (e.g. Physics-informed deep neural networks for learning parameters and constitutive relationships in subsurface flow problems). Both of these ideas are much better than directly differencing the noisy data. The authors should compare more models in the experiments, including the several studies mentioned at the end of the section 2 of this paper, and some other studies dedicated to sparse and noisy data. It is not fair or sufficient to only compare PDE-FIND with the proposed model since it is designed for HR data. The authors should consider more baseline models in the experiments.

### Other concerns
Besides, I am also concerned with the claims on knowledge, data and experimental settings:
* The authors claimed that “Notably, this network is characterized with the capability to encode given physics knowledge”; “The overall network architecture mimics the forward Euler time stepping”; “The recurrent Π-block is a major innovation of this network architecture to capture the dynamics”. What is the difference between PeRCNN and PDE-net and PDE-net 2.0 (using the kernel to represent the derivative term, and use the δt-block to mimic the forward Euler time stepping)?
* The author mentioned that “The obtained coefficients from sparse regression may not fully exploit all the available measurement as the regression is performed on subsampled data.”. Sparse regression is based on HR data, why is it subsampled? Why not use all HR data directly in the sparse regression, but perform additional fine-tuning? If the essence of fine-tuning is coefficient fitting, then the only influencing factor should be the amount of data.
* The author mentioned that “the dominant reconstruction part lacks explicit physical interpretability (i.e., grey-box model), which may induce the difficulty in optimization especially for high-dimensional PDE systems with delicate patterns.”. Why does the process of generating HR data need to be interpretable? It seems that the PDE discovery can be realized as long as the process of mining PDE from HR data is interpretable.
* The author should use the same candidate set in different PDE missions. It seems that the current candidate set is designed according to the PDE to be discovered. Besides, the number of candidates is relatively small, which reduces the difficulty of the problem. In practical applications, it is difficult to give a specific candidate set for a certain mission since we do not know the form of the PDE in advance. The candidate set should contain all possible candidates, and should not change according to the mission.

### Minor points:
There are two “the” repeated at the end of the ninth line on the third page.

### References
1. Xu, H., Chang, H., & Zhang, D. (2020). DLGA-PDE: Discovery of PDEs with incomplete candidate library via combination of deep learning and genetic algorithm. Journal of Computational Physics, 418, 109584.
2. Maslyaev, M., Hvatov, A., & Kalyuzhnaya, A. (2019, June). Data-driven partial derivative equations discovery with evolutionary approach. In International Conference on Computational Science (pp. 635-641). Springer, Cham.
3. Udrescu, S. M., & Tegmark, M. (2020). AI Feynman: A physics-inspired method for symbolic regression. Science Advances, 6(16), eaay2631.
4. Chen, Y., Luo, Y., Liu, Q., Xu, H., & Zhang, D. (2021). Any equation is a forest: Symbolic genetic algorithm for discovering open-form partial differential equations (SGA-PDE). arXiv preprint arXiv:2106.11927.
5. Luo, Y., Liu, Q., Chen, Y., Hu, W., & Zhu, J. (2021). KO-PDE: Kernel Optimized Discovery of Partial Differential Equations with Varying Coefficients. arXiv preprint arXiv:2106.01078.
6. Long, Z., Lu, Y., Ma, X., & Dong, B. (2018, July). Pde-net: Learning pdes from data. In International Conference on Mac

**Summary Of The Paper:**

This paper proposes a deep convolutional-recurrent network plus a sparse regression model for discovering spatiotemporal PDEs. The three-dimensional PDE experimenta l results are used to validate the effectiveness.

**Summary Of The Review:**

This paper addressed an important problem, PDE-discovery and was clearly written. Some points are new but there are some critical issues on novelty, experiments, knowledge and data which suggests a reject.

---

> ### Author Response · Authors · 2021-11-17
> **Response to Reviewer zdB8 (Part 2)**
>
>
> **Response to other concerns:**
>
> 1. **Compared with PDE-Net/PDE-Net 2.0:** Our method hold similarity compared with PDE-Net/PDE-Net 2.0 since both of them use Conv kernels to represent derivatives and a residual block for time marching. However, one major difference is that the feature maps (or derivatives) in our method are aggregated via an elementwise product operation in $\Pi$-block, while in PDE-Net/PDE-Net 2.0 these feature maps (or derivatives) are aggregated using the weighted sum or symbolic combination. This elementwise product operation makes the $\Pi$-block a universal polynomial approximator (see Appendix Section A) with the following advantages on dealing with the PDE discovery task.
>
> * The nonlinear function $\mathcal{F}$ in the form of multivariate polynomial (e.g., $\mathbf{u}\cdot\nabla u+u^2v$) covers a wide range of well-known dynamical systems, such as Navier-Stokes, reaction-diffusion (RD), Lorenz equations, to name only a few. Since the spatial derivatives can be computed by Conv kernels, a $\Pi$-block with $n$ parallel Conv layers of appropriate filter size is able to represent a polynomial up to the $n^\text{th}$ order.
>
> * The $\Pi$-block is flexible in representing the nonlinear function $\mathcal{F}$. For example, a $\Pi$-block with 2 parallel layers of appropriate filter size ensembles a family of polynomials up to the 2$^\text{nd}$ order (e.g., $u$, $\Delta u$, $uv$, $\mathbf{u}\cdot\nabla u$), with no need to explicitly define the basis.
>
>
> 2. **a. Subsampled HR Data:** The sparse regression is performed on subsampled HR data to avoid the very large number of rows in the library matrix $\mathbf{\Theta(U)}$. For example, it would lead to $8.2\times10^6$ (i.e., $801\times101\times101$) rows in the GS RD case if the original reconstructed HR data is fully used. To perform sparse regression on such a large matrix would be extremely slow or even infeasible due to the computer's memory limit. Therefore, this subsampling strategy is adopted. It is noted that, despite randomly downsampled, the resulting HR data still remains large (e.g., 10\%) and contains sufficiently rich information for PDE discovery. **b. Fine-tuning:** We would like to draw the reviewer's attention that the fine-tuning is performed on the measurement data rather than the subsampled HR metadata. The importance of the fine-tuning step has been discussed in Appendix Section G in the revised paper. We have demonstrated that the additional fine-tuning step using all LR measurement data could effectively improve the accuracy of the coefficients.
>
> 3. **Network Interpretability:** The interpretability is indeed unnecessary to a model for generating the HR metadata. We could simply fit the measurement data using any black-box deep learning model and infer the HR data for sparse regression. However, in this paper we show that the good physical interpretability of a model could facilitate the training (or optimization) process and thus improve the accuracy of generating the HR metadata. This benefit has also been illustrated in our response to the reviewer's Baseline Comparison question. In particular, our method utilizes a multiplicative representation (i.e., $\Pi$-block) of nonlinear functions via the elementwise product operation, instead of the commonly used additive representation (i.e., weighted sum) in deep neural networks. By construction, the $\Pi$-block in our method is able to express any polynomial functions (e.g., $uu_x+vu_y-\Delta u$) exactly up to high-order finite difference error. In some scenarios (e.g., when the kernel size equal to 1 or some Conv kernels are fixed), we could even extract directly an explicit expression from the trained model via some symbolic expansion of the network.
>
> 4. **Candidate Library:** Thanks for reviewer's suggestion. In the revised paper, we unify the candidate set (i.e., 70 candidates) for the case of 2D Burgers' and $\lambda$--$\Omega$ equation. The final result remains unchanged as the sparse regression gives the same PDE form using the enlarged candidate set. However, for the case of 2D GS RD equation, we consider a situation that the prior knowledge on the PDE form is available, i.e., the reaction term in the GS RD equation is in the form of polynomial (actually most RD systems has the polynomial reaction term). It is shown that the prior knowledge could effectively narrow down the search space and remedy the extreme sparsity of the measurement data in this example.
>
> **References:**
>
> [1] Zhao Chen, Yang Liu, and Hao Sun. Physics-informed learning of governing equations from scarce data. Nature communications 2021, 12: 6136.
>
> [2] Hao Xu, Dongxiao Zhang, and Junsheng Zeng. Deep-learning of parametric partial differential equations from sparse and noisy data. Physics of Fluids, 33(3):037132, 2021.

---

> ### Author Response · Authors · 2021-11-17
> **Response to Reviewer zdB8 (Part 1)**
>
> We sincerely thank the reviewer for his/her constructive comments.
>
> **Response to main concerns:**
>
> 1. **Method Clarification:** Our method includes *three key steps*: (1) reconstructing the data, aka., generating HR metadata, via the proposed $\Pi$-block network based on LR noisy data, (2) performing sparse regression for discovery of the PDE structure or closed form based on the HR metadata, and (3) fine tuning the coefficients of the discovered PDE where the PDE terms are inherited and thus encoded in a physics-based network, e.g., reshaping the $\Pi$-block with prescribed terms. Hence, the novelty of our work is *twofold*. **First**, we propose the innovative $\Pi$-block to learn the nonlinear dynamics of a physical system. $\Pi$-block is an efficient and accurate representation of the physical dynamics (i.e., $\mathcal{F}$ in Eq. (1)). In Appendix A, we prove that, by construction, $\Pi$-block is able to express any polynomial functions (e.g., $uu_x+vu_y-\Delta u$) up to the high-order finite difference precision. **Secondly**, owing to the discretization in both spatial and temporal dimensions, our method is able to utilize partial prior physical knowledge to reconstruct the data at high-resolution (HR). For example, if the diffusion term $\Delta u$ is known to exist, a highway Conv connection (with FD stencil as the Conv filter) would be built to account for the known physics, while the $\Pi$-block is designed to learn the complementary unknown dynamics. The boundary conditions, if known *a priori*, can also be encoded through physics-based padding, e.g., [1,2,3,4] padded to be [4,1,2,3,4,1] for the periodic boundary condition. These special modules bring significant benefits for accurate metadata generation based on LR noisy data and show clear superiority over fully connected neural networks (FCNN).
>
> 2. **Baseline Comparison:** This is an excellent suggestion. Following the reviewer's suggestion, we have performed additional experiments using a new baseline -- the FCNN coupled with sparse regression (FCNN+SR), in which a FCNN is used to reconstruct the data for computing partial derivatives based on automatic differentiation while sparse regression is used to discover the PDE. It should be noted that, like many other methods [1, 2] that utilize the FCNN to reconstruct the sparse data, the FCNN+SR approach lacks the capability of explicitly encoding prior physics knowledge. Therefore, the prior knowledge of known PDE terms (e.g., diffusion $\Delta \mathbf{u}$) could not be utilized in the data reconstruction phase. Apart from that, the same problem settings in the paper (e.g., data amount, noise level) are adopted by FCNN+SR. The summary of the result is provided in the table below (Table 1 has also been updated in the revised paper). We observed that FCNN+SR works well when the pattern of the solution is relatively smooth, such as in the 2D $\lambda$--$\Omega$ RD and Burgers' equation cases, where FCNN+SR is able to recover completely (or a large portion of) the governing PDEs for various noise level. However, it struggles in the 2D GS RD case since the FCNN (as a global approximator) has difficulties capturing the local details or sharp propagating wave fronts (see Fig. 2c in the paper). Comparing the performance of different methods on multiple cases (see Table 1 in the revised paper), we observed that our method consistently outperforms FCNN+SR and PDE-FIND in different noise levels.
>
> **Table Metrics of the discovered PDE by FCNN+SR**
>
> | Case | Noise level /% | Precision /% | Recall /% | Relative L2 error /% |
> |---|:---:|:---:|:---:|:---:|
> | Burgers' | 0 | 100 | 100 | 0.37 |
> | Burgers' | 5 | 100 | 100 | 0.43 |
> | Burgers' | 10 | 75.0 | 100 | 1.34 |
> | $\\lambda$-$\\omega$ RD | 0 | 100 | 100 | 5.19 |
> | $\\lambda$-$\\omega$ RD | 5 | 85.7 | 100 | 7.50 |
> | $\\lambda$-$\\omega$ RD | 10 | 85.7 | 100 | 15.85 |
> | GS RD | 0 | 37.5 | 60.0 | 95.14 |
> | GS RD | 5 | 33.3 | 57.1 | 143.55 |
> | GS RD | 10 | 33.3 | 57.1 | 162.98 |
>
> 3. **Difference between the $\Pi$-block and PDE-Net/PDE-Net2.0:** This is an excellent remark. One major difference is that $\Pi$-block aggregates the output of multiple Conv layers via an elementwise product operation while PDE-Net adopts the traditional weighted sum of output and PDE-Net 2.0 uses a symbolic neural network architecture. This multiplicative representation of $\Pi$-block makes it more efficient in expressing the nonlinear PDE terms. For example, two parallel Conv layers with three channels would be sufficient to learn the dynamics of the Burgers' equation (i.e., $\Delta u-uu_x-vu_y$ for the component of $u$). With one layer learns $1$, $-u$ and $-v$ while the other layer learns $\Delta u$, $u_x$ and $u_y$, the output of the elementwise product operation would have three channels corresponding to $\Delta u$, $-uu_x$ and $-vu_y$. We have added a discussion in the Appendix to clarify this aspect (see Appendix Section A in the revised paper).

---

### Official Review · Reviewer_x24u · 2021-11-02

**Correctness:** 3
**Technical Novelty And Significance:** 3
**Empirical Novelty And Significance:** 2
**Recommendation:** 5
**Confidence:** 5

**Details Of Ethics Concerns:**

No concern.

**Main Review:**

The authors make substantial clarification and revisions upon the reviews. Especially, the product operator in the proposed block is indeed expanding PDE-net's capability. Based on these, I raise my score to 5, although still not meeting the acceptance bar. The authors may need more time to revise the manuscript but it has been better than before. The main concern is experiments, which need more time. The presentation also needs time for reconsideration and refining. I raise my confidence from 4 to 5 based on the discussion with the authors.

-------------------------------------------------------------------------

○ Strengths:
1. The ability to reconstruct higher-quality data by available physics data.
2. Promising results over PDE-FIND (Rudy, et al. 2017.) on the selected equations.
3. A very thorough review of prior works.

○ Weaknesses:
1. The novelty is limited. The authors propose to reconstruct higher-fidelity data via a recurrent convolutional network, which is straightforward but not very novel. Many image resolution recovery papers have used similar techniques of Deconv to increase resolution. In Eq. (2), Conv filters are used to process data, while there is no indication of how physics information is encoded. In Eq.(5), the loss is computed by both u and u0. It is not sure how and what physics is encoded into your system as claimed in the introduction. Could you explain?
2. Confusing presentation. Many notations are not clearly stated. For example, on Page 5, the authors write both x˜ and u˜ as measurements, which is confusing. On page 3, x and t are not introduced. From the paper, readers can hardly interpret their meanings. What is u_y if y is not an input variable in Function F? It is confusing to put u_{k+1} and u_t together, as the first one means time step while the latter one means time derivative. Moreover, what is the meaning of the dimensionality of u (1xn)? Which one is the spatial dimension and which one is the temporal dimension? The definitions are not included or are vague.
3. Only one baseline published in 2017 is compared in the experimental section, while many papers within the recent three years also claim to improve the robustness of PDE-FIND and solve the low-quality data problem, such as [1-4]. Thus, this paper’s improvement is not persuasive enough without comparisons to state-of-the-art models.
4. This method solely discusses the discovery of PDEs with constant coefficients and discrete data and the selected three equations in the experimental section are quite simple compared to equations used in prior works, whereas many papers in recent years have moved forward to the discovery of more complex PDEs with continuous mesh-free data. Why not experiment on more complex PDEs like prior works do to adapt the real-world scenarios, such as the 2D Navier-Stokes equation? The influence of this work is limited if only discrete data and constant coefficients on simple PDEs are discussed.
5. There is no demonstration of real-world data. The data used in this work is solely based on mesh-based simulation, while most real data do not have high-fidelity at all mesh-grid points, which means the data is not only noisy but also sparse. The authors only discuss the robustness against noise. How well does the method scale with sparsity?
6. The noise range tested in this paper is limited compared to previous work. The authors show that their method is good with noise up to 10%, however, most prior works can also handle 10%. I would be interested to know whether the method can handle larger noise to compete with prior works.

[1] Zhao Chen, Yang Liu, and Hao Sun. Physics-informed learning of governing equations from scarce data. arXiv preprint arXiv:2005.03448, 2020.
[2] Jun Li, Gan Sun, Guoshuai Zhao, and Li-Wei H. Lehman. Robust low-rank discovery of data-driven partial differential equations. In AAAI, pp. 767–774. AAAI Press, 2020a.
[3] Hao Xu, Dongxiao Zhang, and Junsheng Zeng. Deep-learning of parametric partial differential equations from sparse and noisy data. Physics of Fluids, 33(3):037132, 2021.
[4] Valerii Iakovlev, Markus Heinonen, and Harri Lähdesmäki. Learning continuous-time PDEs from sparse data with graph neural networks. In International Conference on Learning Representations, 2021.


**Summary Of The Paper:**

The authors propose a physics-encoded discrete learning framework to improve the robustness of the PDE-FIND algorithm. The physics knowledge such as known terms and initial conditions can be used to reconstruct higher-resolution data. Then, sparse regression is performed with fine-tuning of coefficients to finally determine the discovered PDE.

**Summary Of The Review:**

The paper is technically correct but lacks novelty and clarity. The experimental demonstration is not sufficient to support that it can also work for state-of-the-art PDE discovery models and for more complex PDEs. The reported robustness is also not exceeding the state-of-the-art models. Besides, the presentation and notation definitions can be improved. Based on these observations of the submitted paper, I lean towards rejection.

---

> ### Author Response · Authors · 2021-11-17
> **Response to Reviewer x24u (Part 3)**
>
> 6. **Data Noise:** Thanks for this great comment. To check if our method could handle larger noise, we performed additional experiments with the noise levels of 20\% and 30\% for all three cases. We found that larger noise affects the accuracy of our method differently (see the table below). For the 2D GS RD equation, the accuracy of our method deteriorates drastically as the noise level increases. That is because the resolution of the sparse measurement data used in this example is quite low ($26\times26$). However, for the case of 2D $\lambda$-$\Omega$ RD and Burgers' equations where the measurement data has a decent resolution ($51\times51$), our method stays competitive even for 30\% Gaussian noise. Overall, there is a trade-off between the sparsity and noise level of the measurement data. We have added a discussion on data noise in the Appendix to clarify this aspect (see Appendix Section E in the revised paper).
>
> **Table. Performance of our method under various noise levels**
>
> | Case | Noise level /% | Precision /% | Recall /% | Relative L2 error /% |
> |---|:---:|:---:|:---:|:---:|
> | Burgers' | 0 | 100 | 100 | 0.50 |
> | Burgers' | 5 | 100 | 100 | 0.54 |
> | Burgers' | 10 | 100 | 100 | 0.59 |
> | Burgers' | 20 | 75.0 | 100 | 7.33 |
> | Burgers' | 30 | 75.0 | 100 | 9.10 |
> | $\\lambda$-$\\omega$ RD | 0 | 100 | 100 | 1.18 |
> | $\\lambda$-$\\omega$ RD | 5 | 100 | 100 | 2.69 |
> | $\\lambda$-$\\omega$ RD | 10 | 91.6 | 100 | 5.44 |
> | $\\lambda$-$\\omega$ RD | 20 | 91.6 | 100 | 7.65 |
> | $\\lambda$-$\\omega$ RD | 30 | 91.6 | 100 | 15.89 |
> | GS RD | 0 | 100 | 100 | 1.59 |
> | GS RD | 5 | 100 | 100 | 2.85 |
> | GS RD | 10 | 85.7 | 85.7 | 10.03 |
> | GS RD | 20 | 46.2 | 85.7 | 86.05 |
> | GS RD | 30 | 33.3 | 37.5 | 238.31 |
>
> We hope we had fully addressed the reviewer's comments and concerns. Please do not hesitate to let us know if you have any further questions. Thank you very much.
>
> **References:**
>
> [1] Zhao Chen, Yang Liu, and Hao Sun. Physics-informed learning of governing equations from scarce data. Nature communications 2021, 12: 6136.
>
> [2] Hao Xu, Dongxiao Zhang, and Junsheng Zeng. Deep-learning of parametric partial differential equations from sparse and noisy data. Physics of Fluids, 33(3):037132, 2021.

---

> > ### Comment · Reviewer_x24u · 2021-11-18
> > **Response to authors**
> >
> > I appreciate the authors' efforts. Some problems are now addressed while I still have questions to ask to help evaluate the paper quality.
> >
> > 1. First, regarding the claim of physics-encoded blocks, I wonder if your model is unique to have physics-encoded property. If a boundary condition or a term is known apriori, we can also freeze the term in any sparse regression model and perform auto-differentiation to generate higher-resolution data. I am wondering what precise physics can be encoded in a unique way that prior works cannot do. Second, your clarification suggests that your difference to PDE-net is that you can support multivariate polynomial operations. That is somewhat interesting to expand PDE-net's capability by adding a product operator. Maybe it would be wiser to refer to PDE-net and some earlier works since you did use the same basic Conv idea. By the way, HR labels are not the only necessity for higher resolution generation in the literature.
> >
> > 2. The notation presentations are much clearer now. I appreciate your efforts.
> >
> > 3. The added experiment is to support your novelty in HR. However, my concern, as stated in part 1, is to exclude other models' ability to freeze terms. In practice, it is common that people select terms and freeze them as prior knowledge or to avoid collinearity. What is unique about your block? Also, as stated in part 1, if an important contribution is that you expand PDE-net, maybe it is wiser to compare with PDE-net or at least some state-of-the-art baselines?
> >
> > 4. I suggest 2D Navier-Stokes Equation because people know it as a very complex system in turbulence. GS RD is also complex regarding its wave, but the limit of models can be tested by 1) complex PDE terms, e.g., higher-order terms; 2) parametric PDEs, such as 2D NS Equation with highly nonlinear varying coefficients. Regarding the complex wave, it might be better to choose a PDE with a high Renoid like that of NS, although I am not sure about the Renoid in your GS RD.
> >
> > 5&6. Thanks for the added experiments. It looks like the proposed model can handle some PDEs with bigger noise but not all. I still suggest comparing with some not necessarily all SOTAs at least to convince readers that it is no worse than existing works if a major claim of contribution is to tackle noise. Regarding the sparsity, I may not be precise enough about my words. The current data is obtained by mesh-based simulation (uniform distribution), while we could simulate some schemes where data quality is a challenge without using mesh data. For example, a random sample from the mesh data sparsely can also mimic that. This makes sure that your model works even if the data is not in a mesh.

---

> > > ### Author Response · Authors · 2021-11-19
> > > **Post-response to Reviewer x24u**
> > >
> > > We sincerely appreciate the reviewer's further comments and thoughtful suggestions. Please find our response below.
> > >
> > > **1 and 3.** **Physics-encoded Block:** This is an excellent question. In the sparse regression, all the methods (PDE-FIND, FCNN+SR and ours) freeze the known term (i.e., diffusion) by exempting it from being filtered. Nevertheless, our method is also capable of encoding the *partial physics* (i.e., known terms in PDE) in the time marching scheme to generate the HR metadata while the majority of other NN models (e.g., FCNN, PINN) fail to do so. Though other discrete models (e.g., recurrent ResNet, ConvLSTM) might also have such a capability, we have not seen related works that utilizes partial physics to reconstruct noisy measurement for tackling PDE discovery task. We would like to draw the reviewer's attention that we did not try to claim this aspect as a major contribution, but instead, we intend to demonstrate the incorporation of known physics in a hard-encoding manner facilitate metadata reconstruction. **PDE-Net:** Thanks for the reviewer's suggestion. Indeed, we previously attempted to use to PDE-Net as a data-driven model for baseline comparison. However, it does not perform well on LR measurement data. It should be noted that the original architecture of PDE-Net [1] considers only the high-resolution data (i.e., prediction and measurement has the same spatial resolution). Therefore, we do not think it would be a fair comparison by setting a LR grid for PDE-Net computation. Although possible extension could be made on PDE-Net (e.g., introducing the initial state generator to map LR initial state to HR) to make it a HR predictive model trained on LR data, this exceeds the overarching scope of our work. We sincerely hope the reviewer would understand our consideration. Nonetheless, we have added a description in the revised paper to illustrate the motivation of using multiplicative operation (our $\Pi$-block) rather than additive operation [1] in PDE solution modeling.
> > >
> > >
> > > **4.** The Reynolds number of the Burgers' example is $Re=200$. Following the reviewer's suggestion, we further tested our method on a case with $Re=500$ by decreasing $\nu$ to 0.002. 10\% Gaussian noise is considered in this example. Note that the 2D Burgers' equation is a simplified representation of the NS equation. The snapshots of the reconstructed data and the discovered PDE are presented in Appendix Section I in the revised paper. We can see that the chaotic behavior of the system can be well predicted by our network. The result also shows our method is able to recover the complete PDE formulation, given by:
> > >
> > > $u_t=1.9854\times10^{-3} \Delta u - 1.0072uu_x - 1.006vu_y$
> > >
> > > $v_t=1.9409\times10^{-3} \Delta v - 0.9953uv_x - 1.008vv_y.$
> > >
> > > These equations match the ground truth correctly. We do hope this example would clarify the reviewer's concern.
> > >
> > > **5 and 6.** **Noise Effect:** The primary cause of failure on the 2D GS RD example is the extremely LR measurement data (e.g., $21\times21$ for the system with very rich local patterns). In the experiment, our method could handle up to 30\% Gaussian noise if the measurement is in $51\times51$. regarding the possible comparison with PDE-Net, please refer to our response above. **Mesh Data:** Our method can be used to deal with unstructured measurement data as long as it is subsampled randomly from the mesh. To achieve this, the initial state generator (fully convolutional network) needs to be replaced by a fully connected neural network for generating the HR initial state. The data loss for training the network can be obtained through $\textrm{MSE}\left(\boldsymbol{\widehat{\mathcal{U}}}(\tilde{\mathbf{x}})-\tilde{\mathbf{u}}\right)$ where $\tilde{\mathbf{x}}$ still denotes the set of locations on which measurement data is available. Nevertheless, when the computational domain is irregular, the proposed method needs to be extended by incorporating graph neural network module and graph convolution, which will be studied in the future.
> > >
> > > $$ $$
> > > Again, we'd like to thank the reviewer for the thoughtful comments. We do hope these responses are helpful for the reviewer to re-evaluate our paper. We also sincerely wish the reviewer would consider to increase the rating of our paper. Thank you very much.
> > >
> > > $$ $$
> > > **References:**
> > >
> > > [1] Long, Z., Lu, Y., Ma, X. and Dong, B., 2018, July. Pde-net: Learning pdes from data. In International Conference on Machine Learning, pp. 3208-3216.

---

> > > > ### Comment · Reviewer_x24u · 2021-11-19
> > > > **Response to authors**
> > > >
> > > > Thanks for your responses to help me evaluate your paper better.
> > > >
> > > > 1. Unfortunately, your method that combines freezing apriori terms to generate HR data is not an innovation in my opinion. Prior papers do not mention it because freezing or pre-filtering them is a commonly used preprocessing technique rather than a novel method. People do often use it with auto-differentiation since colinearity and low-rank data force us to freeze some terms apriori. Why is it that other models cannot use apriori terms for generating metadata? The innovation stands out only if the paper presents a better way to encode useful physical information into the system. For example, if we know the equation must adhere to some rules (theoretical laws or experience rules) that no one thought of before, constraining it in the optimization or calculation as the PINN did can be considered as an innovation. That would be different from a preprocessing trick that everyone is using. Or, the empirical or theoretical study on how many apriori terms can have a significant positive result on PDE discovery would also be interesting. Or, the guess or estimation of frozen terms based on some boundary conditions would also be more interesting. The sole demonstration of the benefit of freezing terms for metadata is not surprising, as it is underuse without the worth to mention it specifically.
> > > >
> > > > 3. Regarding the difference to PDE-net, still the major concern is on excluding the ability for PDE-net and other baselines to freeze terms and use auto-differentiation. For example, model A + batchnorm outperforming model B without batchnorm cannot demonstrate that model A is essentially better than model B since batchnorm is a technique that everyone may use and is not new. If the setting is [model A + (proposed new norm that no one use)] or [model A + batchnorm] outperforming model B with batchnorm, the author can then claim that the proposed [model A + new norm] or [model A] is novel and makes a difference. If the authors can show improvement in a more convincing setting, which is critical in this situation, that would be great.
> > > >
> > > > 4. Thank you for adding another experiment to support higher Re. Still, the questions regarding the small candidate set (not very large search space) and not complex PDE structure (no higher-order terms or parametric coefficients) are the main trouble. With this setting, my concern is that other existing models can already accomplish your experiments perfectly. It is hard to tell if the proposed model outperforms the robustness of some well-known models, especially since the added baselines are permitted from using known terms for HR.
> > > >
> > > > 5.6. Although sparsity without mesh is only a suggestion, more baselines (at least two) should be included.
> > > >
> > > > Added strength: The paper proposes a new \Pi-block that supports polynomial and multiplication operations of PDE construction, which is novel. Maybe it is worth discussing more of this in your updated paper.
> > > >
> > > > It should be noted though if the authors otherwise wish to demonstrate the discovery of PDEs that are unknown in the real world or extremely complex indeed as a breakthrough in the science, using reasonable prior knowledge (frozen terms) as side information would not be counted as a bad setting. However, using it for general comparison to baselines is not fair. If the "time marching" property is vital as the authors claim, I suggest at least using FCNN+SR with freezing terms (can without the so-called time marching). I am always looking forward to seeing more convincing results for re-evaluation.

---

> > > > > ### Author Response · Authors · 2021-11-20
> > > > > **Post-response to Reviewer x24u**
> > > > >
> > > > > We sincerely appreciate the reviewer's further comments and suggestions. Please find our response below.
> > > > >
> > > > > **1. Embedding *a priori* PDE Terms:** We completely agree with the reviewer's comment. That is why we mentioned in our previous response: "We would like to draw the reviewer's attention that we did not try to claim this aspect as a major contribution, but instead, we intend to demonstrate the incorporation of known physics in a hard-encoding manner facilitate metadata reconstruction." Although this so-called "trick" module is part of our network, we did not mean this is an innovation. However, the simple, yet effective, way of encoding multiplicative polynomial-type PDE terms into the $\Pi$-block network is the novel strength, as pointed out by the reviewer.
> > > > >
> > > > > **2. PDEs with Variable Coefficients:** The reviewer raised a very challenging, but interesting, question that is worthy to investigate -- discovering **closed-form PDEs with variable coefficients**. Although PDE-Net is capable of reconstructing PDE solutions where variable coefficients are involved, directly finding the **exact closed form** of PDEs with variable coefficients still remains a critical open question. We thank the reviewer for providing insights into our future study.
> > > > >
> > > > > **3. Comparison with PDE-Net:** We appreciate and accept the reviewer's suggestion on adding an additional baseline model (e.g., PDE-Net). The resulting "PDE-Net+SR" (SR denotes Sparse Regression) now serves as the new baseline. In particular, we replaced the $\Pi$-block in our proposed framework by the $\delta t$-block in PDE-Net [1] while keep the rest setup in Fig. 1 (e.g., freezing *a priori* terms and introducing ISG for generating HR initial state) the same. In addition to the frozen terms, 15 Conv kernels of size $5\times5$ serve as the learning component for complementary dynamics. We observe from the results that the $\delta t$-block struggles at recovering the HR data. Interestingly, we observed that the reconstructed data is characterized with checkerboard phenomenon since the network's prediction is only supervised on LR grid (i.e., $26\times26$). To evaluate the performance of PDE-Net quantitatively, we compute the mean squared error of the HR prediction by PDE-Net, which is $8.7\times10^{-3}$ compared with  $4.1\times10^{-5}$ by our method. The final discovered PDE after sparse regression reads $u_t=8.597\times10^{-7}\Delta u$ and $v_t=1.179\times10^{-6}\Delta v$, which contain only the compulsory diffusion term. Due to the additive representation of the $\delta t$-block, theoretically PDE-Net [1] does not possess the capability to exactly express nonlinear terms like $uu_x$ and $u^2v$. To verify this, we observed that the successful applications of PDE-Net in the original paper [1] are limited to linear PDE systems. Therefore, it is not surprising to see PDE-Net has difficulties reconstructing the HR data for the 2D GS RD system whose reaction term is highly nonlinear. We are working on the remaining two cases (i.e., $\lambda$--$\Omega$ RD and Burgers' equation under different noise conditions) and will update the results (Fig. 2 and Table 1) in our revised paper in the two days before Nov 22. We hope this experiment would clarify the reviewer's concern.
> > > > >
> > > > > $$ $$
> > > > > Again, we'd like to thank the reviewer and feel lucky to have all the thoughtful suggestions by the reviewer, which we think are very helpful for improving and completing our paper. Thank you very much.
> > > > >
> > > > > $$ $$
> > > > > **References:**
> > > > >
> > > > > [1] Long, Z., Lu, Y., Ma, X. and Dong, B., 2018, July. PDE-Net: Learning PDEs from data. In International Conference on Machine Learning, pp. 3208-3216.

---

> > > > > > ### Comment · Reviewer_x24u · 2021-11-20
> > > > > > **Reponse to authors**
> > > > > >
> > > > > > 1. The current presentation and claim of physics-encoded property still need revision. Some essential concepts should be reconsidered.
> > > > > >
> > > > > > 2. Several papers have claimed to address this problem (finding the exact closed form of PDEs with variable coefficients). May I kindly provide some (not all) refs [1-4] below? Therefore, it is a critical open problem but is certainly reachable a long time ago. Note that PDE-net [5] also discovers parametric PDEs, although without expressing the exact symbolic form in the presentation. It is easy to judge which terms are useless by looking at the coefficients. Most papers in recent years discover parametric PDE with noisy data, while the authors do not consider parametric PDE at all. It is hard to make me not wonder why the authors do not consider at least some of them.
> > > > > >
> > > > > > 3. Good. An experiment against PDE-Net can show your strength in the multiplication capability, the major novelty. However, is it coming from the "physics-encoded" property or the multiplication capability? A thorough study into the ablation and the importance of each part is very important. Why it works and what is working? How about other sparse regression models (with a full candidate library, unlike PDE-Net that cannot do multiplication)? Do they perform well with freezing terms for HR metadata?
> > > > > >
> > > > > > [1] Rudy, Samuel, et al. "Data-driven identification of parametric partial differential equations." SIAM Journal on Applied Dynamical Systems 18.2 (2019): 643-660.
> > > > > > [2] Xu, Hao, Dongxiao Zhang, and Junsheng Zeng. "Deep-learning of parametric partial differential equations from sparse and noisy data." Physics of Fluids 33.3 (2021): 037132.
> > > > > > [3] Luo, Yingtao, et al. "KO-PDE: Kernel Optimized Discovery of Partial Differential Equations with Varying Coefficients." arXiv preprint arXiv:2106.01078 (2021).
> > > > > > [4] Li, Jun, et al. "Robust low-rank discovery of data-driven partial differential equations." Proceedings of the AAAI Conference on Artificial Intelligence. Vol. 34. No. 01. 2020.
> > > > > > [5] Long, Zichao, et al. "Pde-net: Learning pdes from data." International Conference on Machine Learning. PMLR, 2018.

---

> > > > > > > ### Author Response · Authors · 2021-11-20
> > > > > > > **Response to Reviewer x24u (Part 2)**
> > > > > > >
> > > > > > > **3.** Firstly, we would like to emphasize that "physics-encoded" has the meaning of *threefold*: (1) PDE solution time marching that follows the recurrence of forward Euler integration, (2) encoding *a priori* PDE terms into the $\Pi$-block, and (3) encoding multiplicative polynomial-type PDE terms, or called PDE structure, into the $\Pi$-block. We have added this to the revised paper to make the concept clear. Secondly, the ablation study has been done in the paper (see Appendix Section D). We have demonstrated the effect of embedding *a priori* PDE terms (see appendix Fig. D.1). The reason why the $\Pi$-block network works well for learning nonlinear PDEs is *twofold*: (1) the multiplicative representation (i.e., $\Pi$-block) of nonlinear functions via the elementwise product operation is able to express any polynomial functions (e.g., $uu_x+vu_y-\Delta u$) exactly up to high-order finite difference error, and (2) an explicit expression from the trained model could be constructed via some symbolic expansion of the network. For example, with a few number of filters, a symbolic expansion of the trained $\Pi$-block for the 2D Burgers' equation can be expressed as:
> > > > > > >
> > > > > > > $u_t = 0.0051 \Delta u - 0.95u_x(1.07u - 0.0065v - 0.17) + 0.98u_y(0.0045u - 1.01v + 0.17) + 0.053$
> > > > > > >
> > > > > > > $v_t = 0.0051 \Delta v -0.82v_x(1.22u + 0.0078v - 0.18) - 0.91v_y(0.0063u + 1.08v - 0.17) + 0.058$
> > > > > > >
> > > > > > > It is seen that the equivalent expression of the learned model is close to the genuine governing PDEs, which helps to explain the extraordinary learning capability of our model. These properties guarantee the good approximation performance of the proposed network. We have added a paragraph in the appendix to illustrate this aspect (see Appendix Section J). Thirdly, we have considered the *a priori* terms with freezing in the FCNN+SR baseline model. Unfortunately, it does not work well especially for the 2D GS RD equation (please see our previous response and Table 1 in the paper). Finally, we would like to emphasize that our model works not because of the frozen *a priori* terms but the newly designed architecture.
> > > > > > >
> > > > > > > $$ $$
> > > > > > > Please let us know if the reviewer has further questions. BTW, the revised paper has taken into account the reviewer's above comments and suggestions. We are still working the baseline of PDE-Net to complete Table 1. The final revised version will be uploaded soon.

---

> > > > > > > > ### Comment · Reviewer_x24u · 2021-11-21
> > > > > > > > **Response to authors**
> > > > > > > >
> > > > > > > > Thank you for the emphasized experiments of ablation. The contribution of added product operation in the block is somewhat novel upon the PDE-net. Its capability coupled with apriori terms to generate HR data can certainly work but it needs careful reorganization to explain why prior models with freezing terms to generate HR are not working. More baselines in comparison are pending. Based on the contributions and the current presentation, I raise my score to 5.

---

> > > > > > > > > ### Author Response · Authors · 2021-11-21
> > > > > > > > > **Response to Reviewer x24u**
> > > > > > > > >
> > > > > > > > > >  Its capability coupled with apriori terms to generate HR data can certainly work but it needs careful reorganization to explain why prior models with freezing terms to generate HR are not working. More baselines in comparison are pending.
> > > > > > > > >
> > > > > > > > > **A:** We sincerely thank the reviewer for all the constructive comments and suggestions, patience during the discussion, and consideration to increase the rating of our paper. We will make sure to revise the paper to clarify the aspect pointed by the reviewer above. Once the extra baseline comparison experiment is done (running the codes for totally nine cases would take some time), we will upload the final version of the paper and notify the reviewer in the next 10 hours or so. We hope, by then, the reviewer would have the opportunity to conclude the final evaluation. Thank you very much for your patience.

---

> > > > > > > > > ### Author Response · Authors · 2021-11-22
> > > > > > > > > **Final version of the revised paper uploaded**
> > > > > > > > >
> > > > > > > > > Dear Reviewer x24u:
> > > > > > > > >
> > > > > > > > > We sincerely thank your time and efforts placed on reviewing our paper. We have also enjoyed the fruitful discussions with you. The final version of our revised paper has been uploaded. In particular, the following changes have been made compared with the previously uploaded revision:
> > > > > > > > >
> > > > > > > > > * Figure 2 and Table 1 have been updated, where the results for the new baseline of "PDE-Net+Sparse Regression" have been added;
> > > > > > > > > * The method sections/subsections have been rephrased, where our contributions of the network architecture have been highlighted;
> > > > > > > > > *  Appendices A, E, F, I and J have been revised or added following the reviewer's comments/suggestions.
> > > > > > > > >
> > > > > > > > > Hope all of these help the reviewer make a final evaluation and rating of our paper. Again, thank you very much.
> > > > > > > > >
> > > > > > > > > Best regards,
> > > > > > > > >
> > > > > > > > > The authors of the paper

---

> > > > > > > ### Author Response · Authors · 2021-11-20
> > > > > > > **Response to Reviewer x24u (Part 1)**
> > > > > > >
> > > > > > > We thank the reviewer for the prompt response and also appreciate the fruitful discussions so far. Please see our response below.
> > > > > > >
> > > > > > > **1.** Thanks for this suggestion. We have revised the presentation of the concept of physics-encoded network. The reviewer is referred to the revised version of our paper.
> > > > > > >
> > > > > > > **2a. Parametric PDEs:** Firstly, we would like to draw the reviewer's attention that we are presenting a method in its current form to deal with the problem of "discovering non-parametric PDEs" which still remains a great challenge and keeps drawing significant attention in the community particularly when the measurement data is sparse and noisy. The challenge becomes even worse when the PDE response is very complex (e.g., the GS RD system whose behavior has very rich local patterns with sharp propagating wave fronts). Our **key motivation** is to address these challenges by developing a novel framework, which also sets the scope of our work. As the reviewer knows, there are many types/forms of PDEs, e.g., non-parametric, parametric, stochastic, fractional, integral-coupled, etc. Although the method can be extended for discovering other types of PDEs such as the parametric ones the reviewer suggested, this is out of the scope of our present work. We would prefer to stress our current concentration on discovery of non-parametric PDEs with LR noisy data and leave the reviewer's suggestion to our future study. We hope this clear to the reviewer.
> > > > > > >
> > > > > > > Secondly, we would like to rephrase our previous response to make it more precise: "The reviewer raised a very challenging, but interesting, question that is worthy to investigate -- discovering the closed form of nonlinear PDEs with variable coefficients under **LR noisy measurement data**." The papers listed by the reviewer (some of which have already been cited in our paper) indeed deal with discovery of PDEs with variable coefficients, but either consider low-dimensional PDEs (e.g., 1D) or do not account for measurement scarcity/noise. Discovery of high-dimensional parametric PDEs (e.g., in a 2D/3D spatial domain) with spatiotemporal variable coefficients definitely remains an open and critical challenge. In particular, how to accurately generate the HR metadata based on LR noisy measurements for nonlinear parametric PDE systems remains the key question. We aim to tackle this challenge in our future study (e.g., combining the $\Pi$-block concept and the filter design strategy used by PDE-Net). The rest of sparse regression for PDE formulation discovery could follow a standard routine well studied in the literature, e.g., formulating proper parametric candidate terms.
> > > > > > >
> > > > > > > > From the reviewer: Note that PDE-net [5] also discovers parametric PDEs, although without expressing the exact symbolic form in the presentation. It is easy to judge which terms are useless by looking at the coefficients.
> > > > > > >
> > > > > > > **2b. PDE-Net:** Unfortunately, we do not agree with the reviewer on the statement above. PDE-Net learns the response of (instead of "discovers") linear parametric PDEs through approximation. Although we could manually judge the useless terms of the approximated linear combination, this does not help recover the exact form of PDEs. Again, based on our experiment and the claim where the PDE-Net paper made, the additive operation limits PDE-Net to only learning the response of linear PDEs. Our network could serve as a complementary approach which tackles the challenge of accurate metadata reconstruction for nonlinear PDEs based on LR noisy data.

---

> > > > > > > > ### Comment · Reviewer_x24u · 2021-11-22
> > > > > > > > **A clarification upon some comments of authors**
> > > > > > > >
> > > > > > > > I doubt that authors of PDE-Net will agree with the response of authors that "PDE-Net learns the response of (instead of "discovers") linear parametric PDEs through approximation. Although we could manually judge the useless terms of the approximated linear combination, this does not help recover the exact form of PDEs." I am not an author of PDE-Net, but if you look at the coefficients estimated by PDE-Net, many of them are pure blank, which suggests these terms do not exist. Adding some tricks can easily transform the responses into PDEs. Only in extreme cases where PDE-Net does not find coefficient well, like in some other prior works' studies, PDE-Net can be counted as "not discovering PDEs correctly". PDE-Net at least discusses parametric PDEs. I hardly recall any high-impact papers at the level of good conferences like ICLR that only discuss constant coefficients in mesh-grid data without higher-order terms within the last three years. The complexity of the discussed problem is certainly a metric of consideration for paper quality. It should not be excused by the scope of discussion.

---

> > > > > > > > > ### Author Response · Authors · 2021-11-22
> > > > > > > > > **Response to Reviewer x24u**
> > > > > > > > >
> > > > > > > > > We thank the reviewer for the additional comments. Please see our response below.
> > > > > > > > >
> > > > > > > > > > From the reviewer: I doubt that authors of PDE-Net will agree with the response of authors that PDE-Net learns the response of (instead of "discovers") linear parametric PDEs through approximation. Although we could manually judge the useless terms of the approximated linear combination, this does not help recover the exact form of PDEs. I am not an author of PDE-Net, but if you look at the coefficients estimated by PDE-Net, many of them are pure blank, which suggests these terms do not exist. Adding some tricks can easily transform the responses into PDEs. Only in extreme cases where PDE-Net does not find coefficient well, like in some other prior works' studies, PDE-Net can be counted as "not discovering PDEs correctly".
> > > > > > > > >
> > > > > > > > > **A:** To answer this question, please allow us to first remind the reviewer the definition of *exact form of PDE* which refers to *the parsimonious, closed formulation of partial differential equations in the context of a symbolic expression of a few linear/nonlinear terms*. PDE-Net itself fails to generate such an exact form, although an expressive linear combinational formulation [1] can be realized whose coefficients can even be pruned, e.g.,
> > > > > > > > >
> > > > > > > > > $$\widehat{\boldsymbol{\mathcal{U}}}^{(k+1)} = D_{0}\widehat{\boldsymbol{\mathcal{U}}}^{(k)} + \sum_{1\leq i+j\leq n} \left ( c_{ij}D_{ij}\widehat{\boldsymbol{\mathcal{U}}}^{(k)} \right )$$
> > > > > > > > >
> > > > > > > > > It can be seen from the above equation that when the underlying PDE is linear, the parametric coefficients can be learned/approximated by the filter feature map $D_{ij}$. Despite the resulting formulation is expressive, there is no direct relationship between the learned coefficients and the corresponding symbolic terms of the PDE (e.g., $u$, $u_{x}$, $u_{xx}$, etc.). Pruning does not help uncover the exact form of PDE, which just makes the linear combinational formulation more parsimonious. If the underlying PDE becomes nonlinear (e.g., with terms such as $u^2$, $uu_x$, $u^2v$, etc.), PDE-Net in [1] eventually fails to learn the system response. Hope this clarifies the reviewer's concern.
> > > > > > > > >
> > > > > > > > > > From the reviewer: I hardly recall any high-impact papers at the level of good conferences like ICLR that only discuss constant coefficients in mesh-grid data without higher-order terms within the last three years. The complexity of the discussed problem is certainly a metric of consideration for paper quality. It should not be excused by the scope of discussion.
> > > > > > > > >
> > > > > > > > > **A:** We appreciate and respect the reviewer's opinion. However, we believe this is clearly a **biased comment**. Firstly, no existing papers previously published in high-quality conferences such as ICLR has no correlation to the challenge of the problem itself. More importantly, this should definitely not be considered as a review criterion of ICLR. Many good papers have been published recently in other top avenues, such as Journal of Computational Physics [2], Nature Communications [3] among many others, to tackle the problem of discovering non-parametric PDEs. Secondly, we are uncertain about the reviewer's definition of "high-order" terms. The nonlinear terms in the PDEs we have considered in the paper indeed consist of high-order ones such as $uv^2$, $u^3$, $v^3$, $uv_x$, etc. In particular, the GS RD equation also has huge scale difference in the coefficients (e.g., $10^{-6}$ vs. 1). All of these make the discovery problem itself challenging and complex enough, especially under the condition of LR noisy measurement data. Thirdly, we agree with the reviewer that the discovery of parametric PDEs is an interesting problem. However, we disagree on the reviewer's statement of problem complexity, since this does not reveal the scenario that discovery of parametric PDEs is a more challenging or more complex problem compared with discovery of non-parametric PDEs. It certainly depends on the problem definition (e.g., the complexity of the PDE behavior, the availability of measurement data, etc.).
> > > > > > > > >
> > > > > > > > > **Reference:**
> > > > > > > > >
> > > > > > > > > [1] Long, Z., Lu, Y., Ma, X. and Dong, B., 2018, July. PDE-Net: Learning PDEs from data. In International Conference on Machine Learning, pp. 3208-3216.
> > > > > > > > >
> > > > > > > > > [2] Both, Gert-Jan and Choudhury, Subham and Sens, Pierre and Kusters, Remy. DeepMoD: Deep learning for Model Discovery in noisy data. Journal of Computational Physics 2020, 109985.
> > > > > > > > >
> > > > > > > > > [3] Zhao Chen, Yang Liu, and Hao Sun. Physics-informed learning of governing equations from scarce data. Nature communications 2021, 12: 6136.
> > > > > > > > >
> > > > > > > > >
> > > > > > > > > $$ $$
> > > > > > > > >
> > > > > > > > > It has been a great pleasure to discuss with the reviewer on the open question above. We really appreciate your time as well as your constructive comments/suggestions made through this discussion period. Thank you very much.

---

> > > > > > > > > > ### Comment · Reviewer_x24u · 2021-11-22
> > > > > > > > > > **Response to authors**
> > > > > > > > > >
> > > > > > > > > > Thank you for your efforts in revision. Although your defense does not convince me, I hope my review helps you improve the paper's quality.
> > > > > > > > > >
> > > > > > > > > > -----------------
> > > > > > > > > >
> > > > > > > > > > I want to point out that PDE-Net 2.0 [1] also supports product (multiplication) operators by presenting the exact closed-form PDE (see Table 1 of PDE-Net 2.0), which is why I argue the authors reconsider your manuscript. Clearly, this contradicts what you claimed that PDE-Net cannot discover PDEs.
> > > > > > > > > >
> > > > > > > > > > Many of your methods have strangely high similarities to prior works, without referring to them or discussing the difference. I sincerely hope the authors accept kind advice. I can only consider your proposed block SOMEWHAT novel and the authors should really focus more on discussing the difference between the proposed method and prior works.
> > > > > > > > > >
> > > > > > > > > > [1] Long, Zichao, Yiping Lu, and Bin Dong. "PDE-Net 2.0: Learning PDEs from data with a numeric-symbolic hybrid deep network." Journal of Computational Physics 399 (2019): 108925.

---

> > > > > > > > > > > ### Author Response · Authors · 2021-11-22
> > > > > > > > > > > **Response to Reviewer x24u**
> > > > > > > > > > >
> > > > > > > > > > > Thanks for your prompt reply. We wonder which part of our response does not convince the reviewer. We will be happy to further clarify it.

---

> > > > > > > > > > > > ### Comment · Reviewer_x24u · 2021-11-22
> > > > > > > > > > > > **Response to authors**
> > > > > > > > > > > >
> > > > > > > > > > > > Please kindly refer to the edited response to the authors above.

---

> > > > > > > > > > > > > ### Author Response · Authors · 2021-11-22
> > > > > > > > > > > > > **Response to Reviewer x24u**
> > > > > > > > > > > > >
> > > > > > > > > > > > > > I want to point out that PDE-Net 2.0 [1] also supports product (multiplication) operators by presenting the exact closed-form PDE (see Table 1 of PDE-Net 2.0), which is why I argue the authors reconsider your manuscript. Clearly this contradicts what you claimed that PDE-Net cannot discover PDEs.
> > > > > > > > > > > > >
> > > > > > > > > > > > > **A:** Thanks for your note. The previous comments from the reviewer (the same of our response) have all been referred to PDE-Net [1], rather than PDE-Net 2.0 [2] which just mentioned by the reviewer. We feel a little confused. On a different note, PDE-Net 2.0 represents an excellent work on discovery of non-parametric PDEs, where, however, LR noisy measurement is not considered.
> > > > > > > > > > > > >
> > > > > > > > > > > > > > Many of your methods have strangly high similarities to prior works. I sincerely hope the authors accept kind advice. I can only consider your proposed block SOMEWHAT novel and the authors should really focus more on discussing the difference between the proposed method and prior works.
> > > > > > > > > > > > >
> > > > > > > > > > > > > **A:** We have taken all your constructive comments and suggestions into our paper revision (please also refer to our previous responses to you and the revised paper). We respect, but just hold different opinion toward the final criticism by the reviewer. Nevertheless, we thank the reviewer for your patient discussion.
> > > > > > > > > > > > >
> > > > > > > > > > > > > **References:**
> > > > > > > > > > > > >
> > > > > > > > > > > > > [1] Long, Z., Lu, Y., Ma, X. and Dong, B., 2018, July. PDE-Net: Learning PDEs from data. In International Conference on Machine Learning, pp. 3208-3216.
> > > > > > > > > > > > >
> > > > > > > > > > > > > [2] Long, Zichao, Yiping Lu, and Bin Dong. "PDE-Net 2.0: Learning PDEs from data with a numeric-symbolic hybrid deep network." Journal of Computational Physics 399 (2019): 108925.

---

> > > > > > > > > > > > > > ### Comment · Reviewer_x24u · 2021-11-25
> > > > > > > > > > > > > > **Reponse to authors**
> > > > > > > > > > > > > >
> > > > > > > > > > > > > > Discussion on the difference between PDE-net 2.0 and the proposed block is necessary. It also claims to model P-polynomial. The HR alone does not have enough novelty, with reasons explained before that prior works on metadata and auto-differentiation have discussed the same issue. The main concerns on experiments are not fully resolved, although much better than before. Another suggestion (not related to rate evaluation, does not help improve the novelty) is that it would be much better to explain why pre-filtering can be regarded as prior, why it is reasonable, perhaps relate it to some real cases to better help readers understand and buy in your very basic motivation of raising freezing for HR. (For example, different terms relate to transitional, laminar, inertial, etc. Reasonable to freeze based on knowledge on these.)

---

> > > > > > > > > > > > > > > ### Author Response · Authors · 2021-11-28
> > > > > > > > > > > > > > > **Response to Reviewer x24u**
> > > > > > > > > > > > > > >
> > > > > > > > > > > > > > > Thanks for the reviewer’s additional comments and suggestions, which are constructive and helpful.
> > > > > > > > > > > > > > >
> > > > > > > > > > > > > > > 1. **Difference between our model and PDE-Net 2.0:** This is great suggestion. We will add an Appendix section in the revised paper during the camera-ready phase (if our paper gets accepted) to discuss in detail the difference between our network and PDE-Net/PDE-Net 2.0. Although both our method and PDE-Net 2.0 are capable of modeling polynomial-type PDE forms, several differences are present: (1) PDE-Net 2.0 relies on specified design of the symbolic neural network while our network directly performs multiplication over channels. This will result in much less trainable variables in our network when expressing the some polynomial PDE formulation. (2) When expressing high polynomial order terms (e.g., the 5th order), a deep symbolic neural network (e.g., 5 layers) is needed for PDE-Net 2.0. This adds on difficulty in the network training. However, for our network, we have much better flexibility in expressing high order polynomial PDE formulation (e.g., we can simply add additional feature maps in the $\Pi$-block while keep the layer as one). (3) PDE-Net 2.0 requires careful tuning of the relative weighting for the sparsity regularizer of the symbolic neural network in order to achieve satisfactory accuracy for the $\delta t$-block, due to the nature of the symbolic neural network especially when it is deep, e.g., >2. Our method does not have such an issue.
> > > > > > > > > > > > > > >
> > > > > > > > > > > > > > > 2. **Prior terms:** This is also a great suggestion. The use of frozen *a priori* terms indeed depends on our prior knowledge. For example, in many scientific practices such as biochemical sciences, we know many reaction-diffusion processes have the diffusion part characterized by $\Delta \mathbf{u}$ and the nonlinear reaction part whose closed form formulation remains unknown. In this case, we can assume the diffusion term is known (coefficients unknown) as part of the PDE, which can thus been explicitly embedded in the network through freezing or pre-filtering. We have explained in the paper that this type of embedding is completely *optional*, depending on the prior knowledge we have. We totally agree with the reviewer that this should be discussed clearly in the paper. We plan to add more detailed discussions in the revised paper during the camera-ready phase (if our paper gets accepted).
> > > > > > > > > > > > > > >
> > > > > > > > > > > > > > > We would like to draw the reviewer’s attention that, since the author’s response period has ended, we are not allowed to post the up-to-date version of our revised paper online. Nevertheless, we will make sure to include the above suggestions in our paper. Thank you very much.

---

> > > > > > > > > ### Author Response · Authors · 2021-11-24
> > > > > > > > > **Response to Reviewer x24u: Added an Additional Study on Comparing PDE-Net and Our Method**
> > > > > > > > >
> > > > > > > > > Dear Reviewer x24u: we have performed an additional study on the comparison between PDE-Net and our method. Please see the details below.
> > > > > > > > >
> > > > > > > > > **More experiments on PDE-Net:** We have followed the settings in [1] and slightly adjust the hyperparameters of PDE-Net to ensure its good performance. Due to the additive representation of the $\delta t$-block, theoretically PDE-Net [1] does not possess the capability to exactly express nonlinear terms like $uu_x$ and $u^2v$. We observed that the successful applications of PDE-Net in the original paper [1] are limited to linear PDE systems. Therefore, it is not surprising to see PDE-Net has difficulties reconstructing the HR data for the 2D GS RD system whose reaction term is highly nonlinear. To verify this claim, we performed *two additional experiments* on PDE-Net.
> > > > > > > > >
> > > > > > > > > In the *first experiment*, we use the spatiotemporal high-resolution data of Burgers' equation to train the PDE-Net so that all nodes in the spatiotemporal grid are supervised with labeled data. It turned out that the training error is still large, which verifies that the PDE-Net lacks the expressiveness to fit the *nonlinear response* of Burgers' equation due to its linear expressiveness.
> > > > > > > > >
> > > > > > > > > In the *second experiment*, we consider a linear PDE, i.e., $u_t=0.005\Delta u-u_x-u_y$. We generate the synthetic LR measurement data by downsampling the HR numerical solution. The discretization of domain and the dimension of measurement follow the settings used in Burgers' equation example. The reconstructed HR data by both our method and PDE-Net based on noiseless LR measurement are plotted in the figure (see Google Drive [Link](https://drive.google.com/file/d/1-Ww_F2AGBUQGTn-CIVpE7GIbVwa9cBUW/view?usp=sharing)). We can observe that both PDE-Net and PeRCNN are able to recover the HR solution well when the equation is linear. Furthermore, performing the sparse regression on the reconstructed data yields the correct PDE for both models (see the table below). This example shows that PDE-Net and our PeRCNN model could achieve comparable performance when the PDE system is linear.
> > > > > > > > >
> > > > > > > > > **Table. Performance of PDE-Net+SR and our method on a linear diffusion equation**
> > > > > > > > >
> > > > > > > > >  | Method | Precision /% | Recall /% | Relative L2 error /% |
> > > > > > > > >  |---|:---:|:---:|:---:|
> > > > > > > > >  | PDE-Net | 100 | 100 | 1.2 |
> > > > > > > > >  | Ours | 100 | 100 | 1.1 |
> > > > > > > > >
> > > > > > > > > Since the general discussion period has ended, we were not allowed to revise the paper at this stage. Nevertheless, we will add these results to our paper during the camera-ready phase. We hope this additional experiment could further clarify the reviewer's concern. Thank you very much.
> > > > > > > > >
> > > > > > > > > **Reference:**
> > > > > > > > >
> > > > > > > > > [1] Long, Z., Lu, Y., Ma, X. and Dong, B., 2018, July. PDE-Net: Learning PDEs from data. In International Conference on Machine Learning, pp. 3208-3216.

---

> ### Author Response · Authors · 2021-11-17
> **Response to Reviewer x24u (Part 2)**
>
> 4. **Complex PDE Systems:** Thanks for this comment. We agree with the reviewer that complex PDE systems (like the 2D Navier-Stokes equation mentioned by the reviewer or other PDEs with complex behavior, e.g., chaotic or sharp propagating wave fronts) should be used to test the effectiveness of the proposed method. This is exactly the motivation why we chose the 2D GS RD system as a test example in the paper. We would like to draw the reviewer's attention that the 2D GS RD equation indeed represents a **complex system** with very fine-scale patterns (see the sharp and chaotic propagating fronts shown in Fig. 2c). Furthermore, the scale gap of coefficients in this equation (i.e., ranging from $10^{-6}$ to $0.1$) makes it extremely challenging to uncover the governing PDE from LR noisy data, as a representative case very close to real-world scenarios. To the best of our knowledge, no existing work in the literature has successfully discover the governing equation of such a system given LR data used in our paper. In general, the proposed approach can be naturally used to discover PDEs when snapshot measurements are available (e.g., by camera, laser or other imaging sensors). Moreover, the proposed framework can be naturally extended to deal with mesh-free data, via incorporating graph network modules and graph convolutions. Also, we admit that the current version of the proposed framework is presented in the context of a key assumption that the PDE coefficients are constant, which is commonly seen in real-world applications. We have also clarified this aspect in the revised paper.
>
> 5. **Data Sparsity:** Thanks for this nice comment. We indeed used sparse (LR) data as our measurements. Our method assumes the measurement data is collected on a fixed **coarse** sensor grid (e.g., $21\times21$ sensors), captured either by point-wise sensor units or by imaging techniques. Although the ground truth data is of high fidelity, our measurement data is heavily spatiotemporally-downsampled and, as a result, remains sparse (e.g., a finite number of LR snapshots). As for the scalability with data sparsity, we performed numerical experiments on the 2D GS and $\lambda$--$\Omega$ RD equations using various spatial resolutions, namely, $51\times51$, $26\times26$, $21\times21$ and $11\times11$. The observation is that the tolerable sparsity of our method depends on the spatial pattern of the system. For example, in the 2D GS RD example, the lowest resolution of data to guarantee the discoverability of the governing PDE is $26\times26$ since the very complex maze-like pattern (see Fig. 2c in the paper) is in a relatively fine scale. However, since the solution in the $\lambda$--$\Omega$ RD example is much smoother and periodic (see Fig. 2b in the paper), our method is able to discover a major portion of the PDE (i.e., precision 90.9\%, recall 83.3\%) with the spatial resolution as low as $11\times11$ (a very sparse data scenario). We have added a discussion on data sparsity in the Appendix to clarify this aspect (see Appendix Section F in the revised paper).

---

> ### Author Response · Authors · 2021-11-17
> **Response to Reviewer x24u (Part 1)**
>
> We sincerely thank the reviewer for his/her constructive comments and suggestions, which are very helpful for improving our paper. Please find our responses below. Revisions have also been made in the paper.
>
> 1. **Novelty of the Work:** Thanks for this comment. The novelty of our work is *twofold*. **First**, we propose the innovative $\Pi$-block to learn the nonlinear dynamics of a physical system. $\Pi$-block is an efficient and accurate representation of the physical dynamics (i.e., $\mathcal{F}$ in Eq. (1)). In Appendix A, we prove that, by construction, $\Pi$-block is able to express any polynomial functions (e.g., $uu_x+vu_y-\Delta u$) up to the high-order finite difference precision. **Secondly**, owing to the discretization in both spatial and temporal dimensions, our method is able to utilize partial prior physical knowledge to reconstruct the data at high-resolution (HR). For example, if the diffusion term $\Delta u$ is known to exist, a highway Conv connection (with FD stencil as the Conv filter) would be built to account for the known physics, while the $\Pi$-block is designed to learn the complementary unknown dynamics. The boundary conditions, if known *a priori*, can also be encoded through physics-based padding, e.g., [1,2,3,4] padded to be [4,1,2,3,4,1] for the periodic boundary condition. The loss function is calculated from the prediction and the low-resolution (LR) measurement data, regularized by the initial state generator loss. It is noted that one salient distinction of our work from other existing works on image resolution recovery is that our network does not require any HR labeled data for training. The simple, yet effective, network architecture remains distinct from other existing neural structures (e.g., fully connected or Conv/Deconv neural networks).
>
> 2. **Presentation:** Thanks for reviewer's careful reading. Here, $\tilde{\mathbf{x}}$ denotes the set of locations (i.e., coarse grid) on which the LR measurement data is collected. $\tilde{\mathbf{u}}$ denotes the LR measurement data collected on $\tilde{\mathbf{x}}$. $\boldsymbol{\widehat{\mathcal{U}}}(\tilde{\mathbf{x}})$ denotes the mapping of HR prediction (or reconstructed data) $\boldsymbol{\widehat{\mathcal{U}}}$ on the coarse grid $\tilde{\mathbf{x}}$, i.e., downsampled HR prediction. To avoid the confusion between $u_{k+1}$ and $u_t$, we use the $u^{(k+1)}$ to denote the solution at $k+1$ time step in the revised paper. Also, more definitions are included to improve the readability in the revised paper.
>
> 3. **Baseline Comparison:** This is an excellent suggestion. We have performed additional experiments using a new baseline -- the fully connected neural network coupled with sparse regression (FCNN+SR), in which a fully connected neural network (FCNN) is used to reconstruct the data for computing partial derivatives based on automatic differentiation while sparse regression is used to discover the PDE. It should be noted that, like many other methods [1, 2] that utilize the FCNN to reconstruct the sparse data, the FCNN+SR approach lacks the capability of explicitly encoding prior physics knowledge. Therefore, the prior knowledge of known PDE terms (e.g., diffusion $\Delta \mathbf{u}$) could not be utilized in the data reconstruction phase. Apart from that, the same problem settings in the paper (e.g., data amount, noise level) are adopted by FCNN+SR. The summary of the result is provided in the table below (Table 1 has also been updated in the revised paper). We observed that FCNN+SR works well when the pattern of the solution is relatively smooth, such as in the 2D $\lambda$--$\Omega$ RD and Burgers' equation cases, where FCNN+SR is able to recover completely (or a large portion of) the governing PDEs for various noise level. However, it struggles in the 2D GS RD case since the FCNN (as a global approximator) has difficulties capturing the local details or sharp propagating wave fronts (see Fig. 2c in the paper). Comparing the performance of different methods on multiple cases (see Table 1 in the revised paper), we observed that our method consistently outperforms FCNN+SR and PDE-FIND in different noise levels.
>
> **Table. Metrics of the discovered PDE by FCNN+SR**
>
> | Case | Noise level /% | Precision /% | Recall /% | Relative L2 error /% |
> |---|:---:|:---:|:---:|:---:|
> | Burgers' | 0 | 100 | 100 | 0.37 |
> | Burgers' | 5 | 100 | 100 | 0.43 |
> | Burgers' | 10 | 75.0 | 100 | 1.34 |
> | $\\lambda$-$\\omega$ RD | 0 | 100 | 100 | 5.19 |
> | $\\lambda$-$\\omega$ RD | 5 | 85.7 | 100 | 7.50 |
> | $\\lambda$-$\\omega$ RD | 10 | 85.7 | 100 | 15.85 |
> | GS RD | 0 | 37.5 | 60.0 | 95.14 |
> | GS RD | 5 | 33.3 | 57.1 | 143.55 |
> | GS RD | 10 | 33.3 | 57.1 | 162.98 |

---

### Official Review · Reviewer_t1Ji · 2021-11-02

**Correctness:** 3
**Technical Novelty And Significance:** 3
**Empirical Novelty And Significance:** 3
**Recommendation:** 6
**Confidence:** 2

**Main Review:**

Strengths
1] Procedure is well-defined/described*; Clean figure 2. The method makes sense to me.

Suggestions


1] At first reading, I had a hard time parsing the 2nd contribution bullet point. I was thinking "Is sparse regression the contribution? The inheritance?  Fine-tuning? Or the actual progression through these stages?" After reading, I see what you mean, although instead of 'inheritance' maybe you should say simply how you put it later in the paper "In the fine-tuning step... measurements are used to train a recurrent block completely based on the identified PDE structure from the sparse regression."
* The other thing that took me a second to understand was the recurrent block fine-tuning. I think I understand now that you are scrolling through time computing u_k+1 from u_k using the block from Fig 3b. It may be useful to put the Fine-Tuning loss function -- is it the full vector ||u - U(initial)|| ? Or tuning at each time step before going to the next?

2] I suggest to boldface the winning values in each column of Table 1.

3] The following notes are why I scored a value of 3 on "correctness of claims" and "empirical significance":
When I see such a huge improvement over the baseline, ...
a) I wonder if there is a better comparator worth considering? Unfortunately I am not well versed enough to suggest something specific.
b) I would like to see a progression of difficulty starting from when the methods do approximately the same (e.g., HR non-scarce data) to when a winner emerges (LR + scarce)

**Summary Of The Paper:**

Problem: data-driven PDE discovery methods are not robus to low-quality measurement data.

Solution: A novel architecture that can encode prior knowledge (known terms, PDE structure, boundary conditions), and sparse regression procedure that is hypothesized to be more robust to scarce and noisy data scenarios.

Other Contributions:
* proof that proposed Pi-block is a universal polynomial approximator. This is why I scored a "4" on technical contributions.

**Summary Of The Review:**

The paper presents a clearly described and sensible approach to PDE discovery. The main problem they seek to solve is robustness to data quality and scarcity, which they address with a combination of sparse regularization, fine-tuning, and prior knowledge encoding architecture. I am not extremely familiar with the field but I believe, especially the latter technique, is quite novel. My only real complaint is that only simulation data was used, and it's not "stress tested" in the experiments or against competitors in terms of practical/empirical significance (and hence fully supporting the claims of robustness). Note: I scored myself as a 3 because I'm not that familiar with the related literature.


Edit: My confidence on this topic is low because I do not know the literature a priori. However, I am not convinced by Reviewers x24u, zdB8 that the Pi-block is redundant with PDE-Net (to me it seems more efficient and sparse, at least), and I call into question Reviewer zdB8's claim that this is a "more complex" system than DLGA-PDE. All in all the other reviewers have shaken my confidence in the empirical results and rigor is methodology descriptions, but not as much in the novelty. I will lower my score to 6 to reflect this.

---

> ### Author Response · Authors · 2021-11-17
> **Response to Reviewer t1Ji (Part 2)**
>
> 3.a. **Considering Better Baseline Comparison.** This is an excellent suggestion. We have performed additional experiments using a new baseline -- the fully connected neural network coupled with sparse regression (FCNN+SR), in which a fully connected neural network (FCNN) is used to reconstruct the data for computing partial derivatives based on automatic differentiation while sparse regression is used to discover the PDE. It should be noted that, like many other methods [1, 2] that utilize the FCNN to reconstruct the sparse data, the FCNN+SR approach lacks the capability of explicitly encoding prior physics knowledge. Therefore, the prior knowledge of known PDE terms (e.g., diffusion $\Delta \mathbf{u}$) could not be utilized in the data reconstruction phase. Apart from that, the same problem settings in the paper (e.g., data amount, noise level) are adopted by FCNN+SR. The summary of the result is provided in the table below (Table 1 has also been updated in the revised paper). We observed that FCNN+SR works well when the pattern of the solution is relatively smooth, such as in the 2D $\lambda$-$\Omega$ RD and Burgers' equation cases, where FCNN+SR is able to recover completely (or a large portion of) the governing PDEs for various noise level. However, it struggles in the 2D GS RD case since the FCNN (as a global approximator) has difficulties capturing the local details or sharp propagating wave fronts (see Fig. 2c in the paper). Comparing the performance of different methods on multiple cases (see Table 1 in the revised paper), we observed that our method consistently outperforms FCNN+SR and PDE-FIND in different noise levels.
>
> **Table. Metrics of the discovered PDE by FCNN+SR**
>
> | Case                    | Noise level /% | Precision /% | Recall /% | Relative L2 error /% |
> |-------------------------|:--------------:|:------------:|:---------:|:--------------------:|
> | Burgers'                |        0       |      100     |    100    |         0.37         |
> | Burgers'                |        5       |      100     |    100    |         0.43         |
> | Burgers'                |       10       |     75.0     |    100    |         1.34         |
> | $\\lambda$-$\\omega$ RD |        0       |      100     |    100    |         5.19         |
> | $\\lambda$-$\\omega$ RD |        5       |     85.7     |    100    |         7.50         |
> | $\\lambda$-$\\omega$ RD |       10       |     85.7     |    100    |         15.85        |
> | GS RD                   |        0       |     37.5     |    60.0   |         95.14        |
> | GS RD                   |        5       |     33.3     |    57.1   |        143.55        |
> | GS RD                   |       10       |     33.3     |    57.1   |        162.98        |
>
>
>
> 3.b. **Progression Experiments.** In this paper, we gradually increase the difficulties of the problem by incorporating more Gaussian noise in the measurement data (i.e., from noise-free to 5\% and 10\% noise). The performance comparison given in Table 1 shows that when the measurement data is noise-free, all methods are able to discover the Burgers' and $\lambda$--$\Omega$ equations from data correctly (except the GS RD equation). However, as the noise level increases, the performance of baselines deteriorates drastically or failed (especially for GS RD system) while our model demonstrates robustness against noise.
>
> We hope we had fully addressed the reviewer's comments and suggestions. Please do feel free to let us know if you have any further questions. Thank you very much.
>
> **References:**
>
> [1] Zhao Chen, Yang Liu, and Hao Sun. Physics-informed learning of governing equations from scarce data. Nature communications 2021, 12: 6136.
>
> [2] Hao Xu, Dongxiao Zhang, and Junsheng Zeng. Deep-learning of parametric partial differential equations from sparse and noisy data. Physics of Fluids, 33(3):037132, 2021.

---

> ### Author Response · Authors · 2021-11-17
> **Response to Reviewer t1Ji (Part 1)**
>
> We sincerely thank the reviewer for his/her encouraging and positive feedback. Please find our responses below. Revisions have also been made in the paper.
>
> 1. This is an excellent comment. Our method includes **three key steps**: (1) reconstructing the data, aka., generating HR metadata, via the proposed $\Pi$-block network based on LR noisy data, (2) performing sparse regression for discovery of the PDE structure or closed form based on the HR metadata, and (3) fine tuning the  coefficients of the discovered PDE where the PDE terms are inherited and thus encoded in a physics-based network, e.g., reshaping the $\Pi$-block with prescribed terms. As for the fine tuning loss function, it is the same as the one used in the data reconstruction phase (i.e., Eq. (5)). That said, the fine tuning step optimizes the PDE coefficients based on the error of multiple time steps. The only difference from the data reconstruction phase is that the recurrent network (e.g., the network block) used in the fine tuning is built completely on the discovered PDE structure.
>
> 2. Thanks for this great suggestion. We have boldfaced the winning values in each column of Table 1 in the revised paper.

---

### Official Review · Reviewer_53sM · 2021-11-03

**Correctness:** 3
**Technical Novelty And Significance:** 2
**Empirical Novelty And Significance:** 2
**Recommendation:** 5
**Confidence:** 4

**Main Review:**

- 1. Consider defining all terms in equation (1). Here, x, t, u_x, u_y. Also, notation such as \nabla^2 should be defined in a separate section (in the Appendix) and cited early on (nabla is first defined in section 4.1). Also, the acronym I/BCs is not defined.

- 2. The product block (PB) seems to be central to the discussion but this block is not motivated in the manuscript. Why did the authors choose this architecture? What is the role of PB?
	- a. While eq. (2) utilizes u_k to form \mathcal{F}_hat(u_k), Fig. 1b shows mapping of u_k to u_k. This is pretty confusing. This can be mitigated by labeling 1b according to (2). In other words, where is (2) in Fig. 1b?
	- b. It seems that \mathcal{F}_hat(.) is this supposed to mimic action of \mathcal{F}(.), however it is unclear why it should take this form.
	- c. Essentially, it seems that Fig. 1b is trying to emulate the forward Euler time stepping, as opposed to this being the action of the overall network, since ISG generates HR from LR. If so, consider correcting para. 2 on page 4.
	- d. The terminology "highway physics-based Conv" layers needs citation.

- 3. In Section 3.3:
	- a.  u is defined as a n_s x n_t matrix while in in equation (1) it is a vector of size n. This is pretty confusing since the discussion so far considers u as a vector. Consider updating the notation in (1) by stating the more general spatiotemporal problem.
	- b. Regarding the problem stated in (4) --
		- i) Why do the authors solve (4) via STRidge, when they can use Iterative Hard Threasholding (IHT) (Haupt and Nowak (2006), Blumensath and Davies (2009)) directly? Arguably, there is no need to solve the l2-regularized least squares problem first, which may lead to faster solutions. This choice needs to be justified adequately.
         - ii) Also, (4) is the l0-regularized least squares problem. There is a rich body of literature devoted to l0-minimization, and needs citations to some iconic works, and potentially a section of related works devoted to it (in the Appendix).
		- iii) The paper claims that STRidge "iteratively searches for the optimal tolerance based on the selection criteria (4)". STRidge provides a sparse estimate for a given tolerance, and does not conduct a search. It may be that the authors conducted a search over various choices of the tolerances. If so, consider correcting this statement.

- 4. In Section 3.4:
	- a. u_tilde is a 3d tensor of dimension here. It is unclear why this is a tensor. Consider clarifying and also, define H' and W'.
	- b. The discussion in "Data reconstruction" is very abrupt and does not tie-in with the discussion so far. This is exacerbated by the changing dimensions of u. How does this achieve interpolation? This weakens the entire discussion, so I highly recommend re-writing this part keeping the notation consistent.
	- c. What is an "IC" regularizer? "IC" is not defined.
	- d. What are u_0 and u_tilde?
	- e. Does the sparse regression use the reconstructed HR observations? If so, should this use the italicized capital U notation?
	- f. Last para. pg. 5 -- "The obtained coefficients from sparse regression may not fully exploit all the available measurement as the regression is performed on subsampled data." -- In the discussion preceding this statement the sparse regression was performed on the HR observations.
		- i) Why and when was the data sub-sampled?
		- ii) Why do the authors initially form the HR data when the Sparse regression is performed on the subsampled data?
		- iii) Also, this may be partly due to the use of hard thresholding on the l2-regularized least squares. The l2-regularization may not let the coefficients to scale properly.

	- e. How are these components trained? Although individual components are described, there is no discussion on how these work together or the training procedure. I am assuming these components are trained separately? Consider adding a discussion about this.
	- f. The role of recurrent network in 1b is unclear. The various vertically arranged conv blocks result from the final u_k? Consider clarifying this.

- 5. The authors claim that one of the advantages of the work is its interpretability. Is it due to the interpretability due to the sparse coefficients being able to identify the components of the pde? If yes, then:
	- a. there should be a discussion of the interpretability properties.
	- b. the experiments should demonstrate how the recovered coefficients and the

6. I recommend adding a baseline solving l0-regularized least squared directly using IHT by removing the l2 regularization. This is essential to analyze the performance of the proposed method.

References:
- HAUPT, J. and NOWAK, R. (2006). Signal reconstruction from noisy random projections. IEEE Transactions on Information Theory, 52 4036–4048
- BLUMENSATH, T. and DAVIES, M. E. (2009). Iterative hard thresholding for compressed sensing.
Applied and Computational Harmonic Analysis, 27 265–274

**Summary Of The Paper:**

The paper focuses on discovering the underlying physics dynamics, in the form of partial differential equation (PDE) from low-resolution (LR) measurements. It leverages sparse regression-based recovery to identify the constituents of the unknown PDE. The main contribution is a module that constructs high-resolution (HR) observations from LR data, for sparse regression to perform well.

**Summary Of The Review:**

Although the paper proposes these modules and tries to solve an interesting problem, it is unclear how these modules work together. Also, they use a sparse regression method (STRidge) which seems like a round-about way of solving the problem directly, which may potentially be faster. The paper is well-written in parts with some notational issues and incomplete description. That being said, I hope the comments are useful for updating the manuscript in the response period.

---

> ### Author Response · Authors · 2021-11-17
> **Response to Reviewer 53sM (Part 2)**
>
> 4.a&d). Thanks for the detailed comments. $\mathbf{\tilde{u}}$ (or u\_tilde) is the low-resolution measurement data. It is a 3D tensor because the measurement data is collected on a $H'\times W'$ coarse grid and at $n_t'$ time steps. If the state variable $\mathbf{u}$ has multiple components (e.g., $u$ and $v$), then u\_tilde would be a 4D tensor. The subscript 0 indicates the first snapshot of the measurement data. To avoid the confusion with partial derivative $u_t$, we use the superscript (i.e., $\mathbf{\tilde{u}}^{(0)}$) to denote the time step in the revised paper. Note that $\mathbf{u}$ in Eq. (1) represents the symbol of state variable.
>
> 4.b). Thanks for your comment. In this section, we follow the three stages adopted to discover the governing PDE, e.g., HR metadata reconstruction, sparse regression FOR PDE structure discovery and fine tuning of PDE coefficients. We added more definitions in this subsection (data reconstruction) to ensure the readability.
>
> 4.c). IC is the acronym of initial condition.
>
> 4.e&f). Yes, the sparse regression is performed on subsampled HR (or reconstructed) data to avoid the very large number of rows in the library matrix $\mathbf{\Theta(U)}$. For example, it would lead to $8.2\times10^6$ (i.e., $801\times101\times101$) rows in the GS RD case if the original reconstructed HR data is fully used. To perform sparse regression on such a large matrix would be extremely slow or even infeasible due to the computer's memory limit. Therefore, this subsampling strategy is adopted. It is noted that, despite randomly downsampled, the resulting HR data still remains large (e.g., 10\%) and contains sufficiently rich information for PDE discovery. Such a subsampling strategy is adopted in many papers including the PDE-FIND. To clarify, the partial derivatives involved in the library matrix are computed based on HR data. The subsampling is performed on the rows of library matrix $\mathbf{\Theta(U)}$. As for the notation in the sparse regression, we use the $\mathbf{\Theta(U)}$ (a column vector) to denote the flattened state variable while the HR prediction (italicized capital U) is defined on a spatiotemporal grid, i.e., its a 3D tensor.
>
> 4.g). The initial state generator (ISG) is first pretrained to ensure the accuracy of the input (i.e., $\widehat{\mathcal{U}}^{(0)}$) to the recurrent computation. Then the whole network (ISG and recurrent $\Pi$-block) is trained using the loss function of Eq. (5).
>
> 4.h). We updated the diagram of Fig. 1 in the revised paper. The intermediate output $\mathcal{F}(\widehat{\mathcal{U}}^{(k)})$ from the elementwise product operation is added. The recurrent block (without highway finite difference Conv layer) mimics the time marching $\widehat{\mathcal{U}}^{(k+1)} = \widehat{\mathcal{U}}^{(k)} + \mathcal{F}(\widehat{\mathcal{U}}^{(k)})\cdot\delta t$.
>
> 5). The interpretability of our model lies in that it is possible to extract an explicit expression from the trained model (in data reconstruction phase) via some symbolic expansion of the network. This is because our method utilizes a multiplicative representation (i.e., $\Pi$-block) of nonlinear functions via the elementwise product operation. Therefore, we could multiply out an explicit expression from the learned model. Let us use an example of $\Pi$-block with two parallel $1\times1$ layers, each has two channels. If the input to the $\Pi$-block has the components of $u$ and $v$, the output of the elementwise product operation would be a second order polynomial, i.e., $\mathcal{F}=(w_{00}u+w_{10}v+b_0)(w_{01}u+w_{11}v+b_1)$ where $b_j$ is the bias and $w_{ij}$ is the weight corresponding to $i$ th input channel and $j$ output channel. Similar symbolic computation could be performed if the partial derivatives are involved to obtain an explicit expression for learned $\mathcal{F}$.
>
> We hope we had fully addressed the reviewer's comments and concerns. Please do not hesitate to let us know if you have any further questions. Thank you very much.

---

> > ### Comment · Reviewer_53sM · 2021-11-21
> > **Thank you for the clarifications; a note on IHT variants and speed-ups**
> >
> > Thank you for the reminder and for updating the manuscript. I have read your response and the updated manuscript, it certainly is in a better shape.
> >
> > - Re: IHT( 3.b. & 6): There are two flavors of IHT, while i) Blumensath et. al. keep the top k entries (where k is assumed to be the sparsity), ii) Haupt et. al. threshold the magnitude of the entries iteratively (see Haupt et. al. Pg. 10). I recommended these two to enable the authors to choose the one which fits their case. It seems that the authors followed the algorithm in i) to arrive at the that IHT does not yield good results. Although I appreciate authors' efforts to implement this technique, I am not fully convinced that this means that IHT does not work, since i) requires getting the sparsity right, which in general is difficult. I do think that ii) fits this setting better, and may perform better since it tackles the actual L0 problem based on a chosen threshold. I understand that this is the end of the discussion period, but any evaluation is much appreciated.
> >
> > - Re: Literature Review (3.ii) I don't see any citations/discussion on sparse inference. I highly recommend this addition.
> >
> > Other comments for speed-ups:
> >
> > - Re: Speed-ups (4.e-f) -- Since sparse regression for different snapshots are independent, authors can consider parallelizing them across workers. Or does the architecture require sequential processing?
> >
> > Thanks again.

---

> > > ### Author Response · Authors · 2021-11-22
> > > **Response to Reviewer 53sM**
> > >
> > > > Thank you for the reminder and for updating the manuscript. I have read your response and the updated manuscript, it certainly is in a better shape.
> > >
> > > We sincerely thank the reviewer for the positive feedback as well as the additional comments and suggestions. We Please find our response below.
> > >
> > > 1. **The variant of IHT:** Thanks for the insightful comment and suggestion on possible adoption of IHT. We would like to clarify that the inspiration of using STRidge comes from PDE-FIND [1] where the authors of this paper claimed STRidge is empirically better than many other common $l_0$ optimization algorithms (e.g., LASSO, STLS [2]) after extensive trials. Second, the main idea of STRidge shares large similarity with that of IHT, where iterative hard-thresholding is applied. Thirdly, our main contribution lies in proposing a novel computational framework for PDE discovery, especially the combination with $\Pi$-block network for HR metadata reconstruction based on LR noisy measurements. Meanwhile, any other effective sparse regression methods, such as IHT suggested by the reviewer, can be plugged into our framework. To be honest, we feel very curious about the variant of IHT in the context of PDE discovery. However, given the hard constraint of deadline during this discussion period, we feel stressed to implement the second IHT algorithm for quantitative comparison. We have added a brief description in Section 3.3 in the revised paper to clarify this aspect. We hope to have the reviewer's understanding.
> > >
> > > 2. **References:** Thank you for this excellent suggestion. We have added the following references:
> > >
> > > * Haupt, J. & Nowak, R (2006). Signal reconstruction from noisy random projections.IEEE Trans-actions on Information Theory, 52(9):4036–4048.
> > >
> > > * Blumensath, T. & Davies, M. E. (2009). Iterative hard thresholding for compressed sensing.Ap-plied and computational harmonic analysis, 27(3):265–274.
> > >
> > > * Rudy, S. H., Brunton, S. L., Proctor, J. L., & Kutz, J. N. (2017). Data-driven discovery of partial differential equations. Science Advances, 3(4), e1602614.
> > >
> > > * Brunton, S. L., Proctor, J. L., & Kutz, J. N. (2016). Discovering governing equations from data by sparse identification of nonlinear dynamical systems. Proceedings of the national academy of sciences, 113(15), 3932-3937.
> > >
> > > 3. **Parallel processing for speed-ups:** Thanks for this comment. Although the snapshots are correlated temporally (i.e., next snapshot depends on the previous), the sparse regression could be parallelized with multiple workers as long as inter-core communication is enabled. The architecture itself does not require sequential processing. The reviewer put forward an interesting direction for our future study -- when huge amount of metadata is generated by the network, how to effectively and efficiently perform sparse regression for PDE discovery. We will investigate it in the future.
> > >
> > > $$ $$
> > >
> > > We hope we had addressed the reviewer's questions. We also very much appreciate if the reviewer would consider to re-evaluate/re-rate our revised paper. Thank you very much.

---

> > > > ### Comment · Reviewer_53sM · 2021-11-22
> > > > **Thank you for the response**
> > > >
> > > > I would like to thank the authors for their quick response, and will reconsider my scores based on the main claimed contribution in the next discussion phase with other reviewers. Although I don't think STRidge is ideal for the task, I agree that this module can be swapped out with other sparse regression techniques. As a result, this will not be used against the work.
> > > >
> > > > Thanks again.

---

> > > > > ### Author Response · Authors · 2021-11-23
> > > > > **Response to Reviewer 53sM**
> > > > >
> > > > > We would like to sincerely thank the reviewer for the positive feedback. Your comments and suggestions have been constructive and helpful for improving our paper. Thank you very much.

---

> ### Author Response · Authors · 2021-11-17
> **Response to Reviewer 53sM (Part 1)**
>
> We sincerely thank the reviewer for his/her detailed and constructive comments and suggestions, which are very helpful for improving our paper. Please find our responses below. Revisions have also been made in the paper.
>
> 1). Here $\mathbf{x}$ and $t$ denote the spatial and temporal coordinate of a system. The subscript on a variable denote the partial derivative (e.g. $\mathbf{u}_x=\partial \mathbf{u}/\partial x$). I/BCs is the acronym for initial and boundary conditions. More explanations are included in the revised paper (see Section 3.1).
>
> 2). Thanks for this comment. Product ($\Pi$) block is crucial to the representation of the nonlinear dynamics function $\mathcal{F}$. As the highway Conv connection (with finite difference stencil as the Conv filter) accounts for the **known physics**, this product block is used to learn the **complementary unknown dynamics**. In Appendix Section A, it is demonstrated that the product block is a universal polynomial approximator with the following advantages on representing the nonlinear function $\mathcal{F}$ in the PDE discovery task:
>
> * The nonlinear function $\mathcal{F}$ in the form of multivariate polynomial (e.g., $\mathbf{u}\cdot\nabla u+u^2v$) covers a wide range of well-known dynamical systems, such as Navier-Stokes, reaction-diffusion (RD), Lorenz equations, to name only a few. Since the spatial derivatives can be computed by Conv kernels, a $\Pi$-block with $n$ parallel Conv layers of appropriate filter size is able to represent a polynomial up to the $n^\text{th}$ order.
>
> * The $\Pi$-block is flexible in representing the nonlinear function $\mathcal{F}$. For example, a $\Pi$-block with 2 parallel layers of appropriate filter size ensembles a family of polynomials up to the 2$^\text{nd}$ order (e.g., $u$, $\Delta u$, $uv$, $\mathbf{u}\cdot\nabla u$), with no need to explicitly define the basis.
>
> In addition, we updated the notations in Fig. 1. Now the superscript is used to denote the time step, e.g., $\widehat{\mathcal{U}}^{(0)}$ is the first snapshot of the HR prediction. The intermediate output $\mathcal{F}(\widehat{\mathcal{U}}^{(k)})$ from the elementwise product operation is added so that the whole recurrent computation (time marching) of $\widehat{\mathcal{U}}^{(k+1)} = \widehat{\mathcal{U}}^{(k)} + \mathcal{F}(\widehat{\mathcal{U}}^{(k)})\cdot\delta t$ is clear.
>
> 3.a). Thanks for pointing out it. To avoid the confusion, in the revised paper we use $u$ to replace the $\mathbf{u}$ for denoting one component of the state variable (e.g., $\mathbf{u}=[u,v]^{\texttt{T}}$) defined on a $n_s\times n_t$ **spatiotemporal grid**. Herein, $n_t$ is the number of time steps and $n_s$ is the number of sensors. To perform the sparse regression, $u$ is flatten into a column vector $\mathbf{U}$ whose shape is $n_s\cdot n_t\times 1$. Accordingly, $\boldsymbol{\Theta}(\mathbf{U})\in \mathbb{R}^{n_s\cdot n_t\times s}$ is the library matrix consisting of $s$ candidate terms.
>
> 3.b & 6). Thanks for the reviewer's suggestion on IHT. We have tested the IHT algorithm on 2D Burgers equations, but the discovered result is worse than that of STRidge. Moreover, the performance of IHT heavily relies on the selection of sparsity term (i.e., pre-defined number of non-zeros values), which should remain unknown *a priori* in our problem setting. For the description of the STRidge algorithm, we have corrected it in the revised paper (see Section 3.3).
>
> Specifically, we have tested the IHT algorithm with different numbers of non-zeros values (i.e., mSparsity), where we can see the discovered PDE structure and L2 error reaches best when mSparsity equals to 4. The ground truth is actually 3 but the L2 error is worse when mSparsity is 3.
>
>  |                |       |       |        |       |       |       |       |       |       |       |
>  |----------------|:-----:|:-----:|:------:|:-----:|:-----:|:-----:|:-----:|:-----:|:-----:|:-----:|
>  | mSparsity      |   1   |   2   |    3   |   4   |   5   |   6   |   7   |   8   |   9   |   10  |
>  | Regression err | 0.707 | 0.884 | 0.344  | 0.029 | 0.079 | 0.291 | 0.100 | 0.147 | 0.188 | 0.222 |
>  |                |       |       |        |       |       |       |       |       |       |       |

---

### Official Review · Reviewer_yjyY · 2021-11-04

**Correctness:** 4
**Technical Novelty And Significance:** 3
**Empirical Novelty And Significance:** Not applicable
**Recommendation:** 6
**Confidence:** 3

**Main Review:**

Major comments:

-) The written English language needs to be improved -- the paper is reasonably written from a technical perspective but
grammar is not on par.

-) In Section 4, how is the ground truth obtained?

-) One thing I find missing is wall-clock times. I can see that for the experiments and parameters reported in Section 4, the
proposed algorithm returns better qualitative results. Nonetheless, how much faster/slower is the new method versus the
competition?

-) My main concern is that the sparse regression step can be very slow due to the very large library \Theta(U). Isn't a pre-processing
step like PCA needed in practice? The reason I mention this is because several columns of this matrix are not contributing much.

-) In terms of novelty, the sparse regression step is fairly well-known. I know of some work done on reconstructing noisy samples
(the authors cite a good part of the related literature on the topic) but how exactly step 1 is really different from previous work?
The authors discuss the topic but it's not really clear.

-) Unfortunately, the fine tuning step depends on that of sparse linear regression, i.e., the correct PDE form need be determined.
The authors mention this themselves, but any guidance on whether the entire mechanism need be restarted (with different parameters)
or some of the work can be salvaged would be interesting.

Minor comments:

-) "Recently, a ground-breaking work of sparse identification of nonlinear Dynamics (SINDy)..." Please capitalize as needed.
-) "Furthermore, the introduction of spatial derivative terms and improved the sparsity-promoting algorithm of sequential threshold ridge regression..." This phrase doesn't make sense.
-) "...many works attempt to relieve the pain..." Please re-phrase.

**Summary Of The Paper:**

The focus of this paper is to develop a learning framework to discover spatiotemporal PDEs from scarce and noisy data.
The authors suggest devise a deep convolutional-recurrent network which encodes prior physics knowledge followed by
sparse regression with reconstructed data to identify the analytical form of the governing PDEs. The framework is validated
on three high-dimensional PDE systems where superiority over baselines is demonstrated.

**Summary Of The Review:**

Overally, the proposed algorithm is discussed at a good abstract level, and experiments on a few small datasets are provided.
See my comments above for more details.

---

> ### Author Response · Authors · 2021-11-17
> **Response to Reviewer yjyY (Part 2)**
>
> **Response to minor comments:**
>
> 7. Thanks for the reviewer's careful reading. We have fixed these minor issues. In particular, we have capitalized the “...sparse identification of nonlinear Dynamics (SINDy)...” as “...Sparse Identification of Nonlinear Dynamics (SINDy)...”. Moreover, we have revised the sentence “Furthermore, the introduction of spatial derivative terms and improved the sparsity-promoting algorithm of sequential threshold ridge regression (STRidge) makes SINDy is applicable to general PDE discovery...” as “Furthermore, the introduction of spatial derivative terms and the improvement of the sparsity-promoting algorithm as Sequential Threshold Ridge regression (STRidge) make SINDy applicable to general PDE discovery...”. Lastly, we have rephrased “...many works attempt to relieve the pain...” as “...many works attempt to solve this issue...”
>
> We hope we had fully addressed the reviewer's comments and concerns. Please do not hesitate to let us know if you have any further questions. Thank you very much.

---

> ### Author Response · Authors · 2021-11-17
> **Response to Reviewer yjyY (Part 1)**
>
> We sincerely thank the reviewer for his/her constructive comments and suggestions, which are very helpful for improving our paper. Please find our responses below. Revisions have also been made in the paper.
>
> **Response to major comments:**
>
> 1. Many thanks for the reviewer's suggestion. We have checked the paper thoroughly and revised the paper, especially on the grammar. Please see the revised paper with track changes (marked in red color).
>
> 2. The ground truth is generated numerically using the finite difference (FD) method. More details on generation of the ground truth and synthetic measurement data can be found in Section 4.3 in the revised paper.
>
> 3. This is a good question. The PDE-FIND reconstructs the noisy data via local interpolation and regression, which leads to negligible time cost. However, the critical bottleneck of PDE-FIND lies in its requirement of large high-quality (clean) structured measurement data, owing to its use of numerical differentiation, which poses critical limitation of PDE-FIND in practical applications where data is sparse and noisy. In contrast, our method uses a deep network to reconstruct the full-field solution based on noisy data (in Step 1 shown in Figure 1) and then to fine tune the coefficients (in Step 3 shown in Figure 1). Therefore, our method requires more wall-clock time to perform the equation discovery, primarily due to the network training process. However, we demonstrate that this time-consuming data reconstruction procedure with our physics-encoded model is crucial when the measurement data is heavily corrupted and sparse. There is obviously a trade-off between computational efficiency and need of high-quality data PDE discovery.
>
> 4. Thanks for this comment. Since the size of the library (i.e., number of candidate functions) usually ranges from dozens to one hundred, the sparse regression is quite fast (\~seconds) thanks to the efficient thresholded Ridge regression performed at each iteration. PCA is not needed because the thresholding operation can promptly remove the columns of the matrix that are not contributing much, iteratively shrinking the size of the matrix during the sparse regression process. Overall, we observe that the sparse regression technique we adopted has a good  computational efficiency, given the fact that the rank of the matrix is relatively small, e.g., $\leq100$.
>
> 5. This is an excellent comment. Like the reviewer mentioned, there exist a variety of methods to reconstruct noisy measurement data, such as B-spline or polynomial interpolation, fully connected neural network (FCNN) and convolutional neural network. Compared with the existing methods, the novelty of our model for data reconstruction lies in two aspects: (i) The discretization in the temporal dimension (i.e., forward Euler time marching) renders our model the capability to encode known terms in PDEs to reconstruct the noisy data. We could simply create a highway connection (e.g., $\Delta u$ for the known diffusion term as part of our *a priori* knowledge) to account for the known physics. (ii) The recurrent $\Pi$-block proposed in our method is an efficient and flexible representation for unknown nonlinear dynamics. It is able to express any polynomial-type PDE functions (e.g., $uu_x+vu_y-\Delta u$) up to the high-order finite difference precision (see Appendix Section A in the paper). In the revised paper, we included the results of a new baseline -- FCNN+SR which uses the FCNN to reconstruct the noisy data and the sparse regression (SR) to discover the form of PDEs. The result clearly shows the superiority of our proposed network for noisy data reconstruction and consequently the PDE discovery.
>
> 6. Thanks for pointing it out. Although the fine tuning step depends on the result of sparse regression, we use a hold-out validation dataset to ensure that the discovered equation generalizes, aka., we evaluate the error of the discovered equation on the validation dataset. If the validation error is much higher than the training error, we would restart the sparse regression process using different parameters. Similar cross validation step is performed on the data reconstruction phase to ensure the trained model generalizes. We have clarified this aspect in the revised paper.

---

### Author Response · Authors · 2021-11-18
**Revised paper uploaded and response to reviewers' comments posted**

Dear Reviewers:

We would like to thank you for your constructive comments and suggestions, which are very helpful for improving our paper. We have posted point-to-point reply to each question/comment raised by you and uploaded the revised version of our paper (with track changes marked in red). Please do feel free to let us know if you have any further questions.

Thank you very much.

Best regards,

*The Authors of the Paper*

---

### Author Response · Authors · 2021-11-29
**Many thanks for the reviewers' constructive comments and suggestions**

Dear Reviewers,

We would like to sincerely thank you for your great efforts and time placed on reviewing our paper. Your comments and suggestions have been very constructive, insightful and helpful for improving the quality of our paper. Special thanks are given to Reviewers **x24u** and **zdB8** for increasing their recommendation scores after detailed discussions.

We are delighted to hear more feedback from you. Please feel free to let us know if you have any additional questions or concerns. Thank you very much.

Best regards,

The authors of the paper

---

### Decision · Program_Chairs · 2022-01-20

**Decision:**

Accept (Poster)

**Comment:**

The paper introduces a pipeline to discover PDEs from scarce and noisy data. Reviewers engaged in a very thoughtful discussion with the authors. I read the extensive rebuttal, and I believe the authors have addressed the major concerns claimed by the reviewers. I ask the authors to make sure to include all the changes and additional experiments in the camera-ready version.